# Towards a more general understanding of the algorithmic utility of recurrent connections

**Brett W. Larsen**[1,2,3], **Shaul Druckmann**[2,3]*

**1** Department of Physics, Stanford University, Stanford, California, United States of America, **2** Department of Neurobiology, Stanford University School of Medicine, Stanford, California, United States of America, **3** Wu Tsai Neurosciences Institute, Stanford University, Stanford, California, United States of America

* shauld@stanford.edu

**Data Availability Statement:** Our code is currently available on Github: 1. Edge-connected pixel task https://github.com/druckmann-lab/edgeConnectedPixel 2. Competitive foraging task

## Abstract

Lateral and recurrent connections are ubiquitous in biological neural circuits. Yet while the strong computational abilities of feedforward networks have been extensively studied, our understanding of the role and advantages of recurrent computations that might explain their prevalence remains an important open challenge. Foundational studies by Minsky and Roelfsema argued that computations that require propagation of global information for local computation to take place would particularly benefit from the sequential, parallel nature of processing in recurrent networks. Such "tag propagation" algorithms perform repeated, local propagation of information and were originally introduced in the context of detecting connectedness, a task that is challenging for feedforward networks. Here, we advance the understanding of the utility of lateral and recurrent computation by first performing a large-scale empirical study of neural architectures for the computation of connectedness to explore feedforward solutions more fully and establish robustly the importance of recurrent architectures. In addition, we highlight a tradeoff between computation time and performance and construct hybrid feedforward/recurrent models that perform well even in the presence of varying computational time limitations. We then generalize tag propagation architectures to propagating multiple interacting tags and demonstrate that these are efficient computational substrates for more general computations of connectedness by introducing and solving an abstracted biologically inspired decision-making task. Our work thus clarifies and expands the set of computational tasks that can be solved efficiently by recurrent computation, yielding hypotheses for structure in population activity that may be present in such tasks.

## Author summary

In striking contrast to the majority of current-day artificial neural network research which primarily focuses on feedforward architectures, biological brains make extensive use of lateral and recurrent connections. This raises the possibility that this difference makes a fundamental contribution to the gap in computational power between real neural circuits and artificial neural networks. Thus, despite the difficulty of making effective comparisons

https://github.com/druckmann-lab/competitiveForaging.

**Funding:** B.W.L. was supported by the Department of Energy Computational Science Graduate Fellowship program (DE-FG02-97ER25308). S.D. is supported by NIH grant R01EB028171 and Simons Collaboration on the Global Brain grant 542969SPI. The funders had no role in study design, data collection and analysis, decision to publish, or preparation of the manuscript.

**Competing interests:** The authors have declared that no competing interests exist.

between different network architectures, developing a more detailed understanding of the computational role played by such connections is a pressing challenge. Here, we leverage the computational capabilities of large-scale machine learning to robustly explore how differences in architectures affect a network's ability to learn tasks that require propagation of global information. We first focus on the task of determining whether two pixels are connected in an image which has an elegant and efficient recurrent solution: propagate a connected label or tag along paths. Inspired by this solution, we show that it can be generalized in many ways, including propagating multiple tags at once and changing the computation performed on the result of the propagation. Strikingly, this simple expansion of the tag propagation network is sufficient to solve a crucial abstraction to temporal connectedness at the core of many decision-making problems, which we illustrate for an abstracted competitive foraging task. Our results shed light on the set of computational tasks that can be solved efficiently by recurrent computation and how these solutions may relate to the structure of neural activity.

## Introduction

One of the brain's most striking anatomical features–and one most divergent from the largely feedforward architectures found in most contemporary artificial neural networks–is the ubiquity of lateral and recurrent connections. Yet our theoretical understanding of the utility of such connections remains limited. In the early days of AI research, Minsky and Papert provided, in their famous discourse on perceptrons, concrete examples of tasks in which feedforward networks of the time were inefficient and suggested that it would take parallel, sequential approaches (akin to recurrent neural networks or RNNs) to solve those problems efficiently [1]. On the other hand, classical results have shown that with enough neurons, a feedforward network can solve any task [2–5], and as hardware and software for training artificial neural networks improved, the tremendous success of deep learning has demonstrated that purely feedforward networks have powerful computational capabilities outperforming most hand engineered approaches [6]. In fact, recent research proposes replacing recurrent architectures even for tasks classically considered natural for recurrent neural networks such as translation [7], further questioning the necessity of recurrent computation. At the same time, a major gap exists between the computational capabilities of artificial agents and those of animals, which are driven by neural circuits. This raises the possibility that the ubiquity of lateral and recurrent connectivity of biological circuits may in part account for the difference between artificial and natural agents. Thus, gaining a more detailed understanding of the computations that are particularly well enabled by lateral and recurrent connections may not only advance our understanding of brain function, but also yield more advanced artificial learning agents.

Minsky and Papert's foundational insight was that computations which are global functions of the input are extremely inefficient when implemented in a single layer perceptron–the state-of-the-art feedforward network at the time–but may not suffer from the same inefficiency when implemented by serial computation (e.g., a recurrent architecture). While the classical example of a global function is the parity function on binary variables, which has a positive output if the number of positive inputs is odd and where changing any one of the inputs changes the output, Minsky and Papert demonstrated more biologically-relevant computations as well. A key example they put forth was connectedness, a computation at the core of object recognition that determines whether two points are connected by pixels of similar color or texture (**Fig 1A**). This computation has a global character since the connectedness property

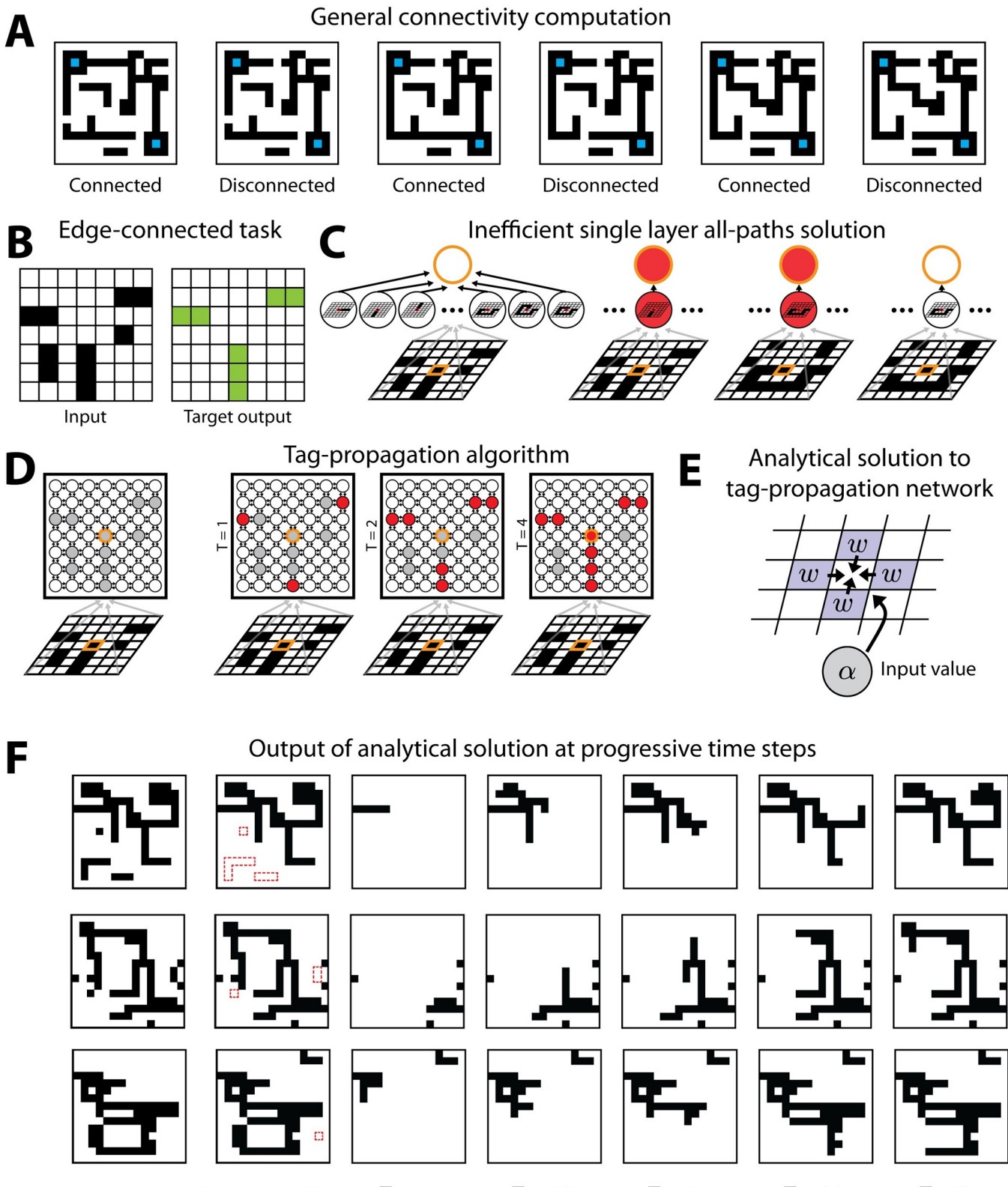

**Fig 1. Edge-connected pixel task.** (A) Illustration of the general connectivity task and its global nature. (B) Example of an input image and correct labelling for that image in the edge-connected pixel task. (C) Implementation of the single-hidden layer architecture for detecting whether the center pixel is connected to

the edge. Each hidden neuron checks whether a certain pattern connecting the pixel to the edge is in the image. (D) Tag propagation solution in a recurrent network for detecting whether the center pixel is connected to the edge. We start with any on pixels connected to the edge tagged as connected. At subsequent time steps, the tag is passed to any neighboring pixel which is also on. (E) Schematic of the setup for the analytical solution of implementation of the tag propagation algorithm in neural network weights. The same setup is repeated at all pixels in the image. (F) Output of the analytical tag-propagation network at progressive time steps for several example images.

between two objects can be altered by toggling one or a few pixels at many locations throughout the image (**Fig 1A**). More formal recurrent network solutions to the problem of calculating connectedness were presented in the late 1990s, when in a series of seminal papers, Roelfsema and colleagues [8–12] demonstrated that sequential, parallel computation networks (very similar to what are currently referred to as "vanilla RNNs") can efficiently establish connectedness through a "tag propagation" process. However, their ability to compare tag propagation within a recurrent network to feedforward solutions, and particularly to deep feedforward networks, was computationally limited because it predated hardware and software improvements in training artificial neural networks [13]. Thus, it remains unclear whether computations that are global functions of the input continue to pose challenges for networks with feedforward architecture.

Here, we first perform a detailed large-scale empirical study to explore the efficiency of a variety of feedforward, recurrent and hybrid neural architectures for the computation of connectedness. We establish that when no constraints are placed on the computation time, recurrent networks' efficiency at detecting connectedness stands up to rigorous architectural search. However, when restrictions on computational time are imposed–a condition likely highly relevant for biological circuits due to the importance of timely actions–fast feedforward architectures can outperform recurrent architectures. Moreover, we demonstrate that combining feedforward and recurrent solutions into hybrid switching networks can be very effective in the presence of temporal constraints while still being able to use extended processing time to improve computation when it is available. Existence of such a mechanism would predict distinct changes in dynamics when increasing limits on reaction time are experimentally imposed. It has been recently shown experimentally that the degree of recurrent computation involved in processing of images depends in a predictable way on their complexity [14]. Here, we further predict that this dependency itself will be modulated by restrictions on reaction time. Additionally, we demonstrate that incorporating partial architectural constraints, i.e., masks that encourage spatial locality, permits artificial neural networks to discover highly effective solutions that are not just approximations of the intuitive recurrent solution.

In an important and complimentary line of work, Linsley et al. [15] (and follow up papers [16–18]) demonstrate the advantages of a particular biologically-inspired network architecture, the hGRU, over feedforward networks in a related task revolving around the detection of long spatial correlations. Furthermore, there is growing evidence for the computational advantage and perhaps even necessity of recurrence in diverse aspects of vision [14, 19–22]. In this work, we focus on thorough characterization of large classes of network architectures (with increasing constraints) with an eye to our generalization of the architecture and tasks. The results of this paper are primarily focused on developing conceptual understanding through abstract tasks which we intend to provide a foundation for important extensions into real-world settings and neural data.

Lastly, we extend the classical notion of spatial connectivity to more abstract connectivity, the connection of a current state to its possible future states, and demonstrate that an extension of the tag propagation architecture can solve a prototypical decision making task that relies on prediction about future environmental states. Combined, these studies further detail how tag-like computations can distill global information into local information and begin to extend our understanding of the types of tasks that may benefit from recurrent connectivity.

## Results

### Computational framework for precisely defined connectedness: Edge-connected pixel task

To establish a computational framework for quantitative comparison of the efficiency of different network architectures at recognizing connectedness, we developed a variant of the general connectedness task where connectedness could be defined in a straightforward and precise manner. Biological visual systems use a variety of features such as color, texture, and more to establish connectedness. To factor out complex questions of processing of texture and color we chose to work with binary images and defined two pixels as being connected if there is a path between them in the image of positive pixels (**Fig 1A**). Similarly, to avoid dependence on specific object geometry, we studied a task whose goal is to label all pixels connected to the edge of the image, which we refer to as the edge-connected pixel task (**Fig 1B**). This can be thought of as performing the connectedness task many times in parallel for all the edge pixels. More specifically, we used binary $N \times N$ images $\boldsymbol{I} = \{-1,+1\}^{N \times N}$ such that the image has $N^2$ pixels and refer to +1 pixels as "on" and -1 pixels as "off." We defined a pixel's connectivity according to its four-facing nearest neighbors, i.e., paths were not considered diagonally between pixels. Thus, we defined the overall edge-connected pixel task as taking a binary image $\boldsymbol{I}$ as input and returning an image that labels all pixels that are on and connected to the edge of the image as +1, i.e., edge-connected or simply connected, and all other pixels as -1, i.e., edge-disconnected or simply disconnected (**Fig 1B**).

As in pixel-pair connectivity, the edge-connected task would be extremely inefficient to solve by enumerating all possible paths since the number of paths grows exponentially with image size (**Fig 1C**). Akin to the solution suggested by Roelfsema for pixel-pair connectivity [8, 9], the task can instead be solved efficiently by a tag-propagation strategy. In an architecture with number of units equal to the number of image pixels in which each unit describes a pixel of the input image, we first set the initial labels to be +1 (connected) for all the edge pixels for which the input is +1 (on) and label the remaining pixels as -1 (disconnected). We then propagate these labels by sequentially setting labels to +1 (connected) if their input is +1 (on) and they have a nearest neighbor that is labeled as +1 (connected). This sequential propagation continues until convergence, i.e., an update does not change the condition of any label (**Fig 1D**).

We first investigated analytically whether the standard formulation of a recurrent network can implement a tag-propagation solution. In the tag propagation approach, we started from an initial set of activated pixels and activated pixels sequentially if they were connected to an active pixel. Assuming as above for simplicity that the recurrent network has as many neurons as pixels, the state of the recurrent network at time $t$, $\mathbf{x}^{(t)}$, is a vector of dimensionality equal to the number of pixels. If as above each neuron corresponds to a specific pixel, i.e., it receives input-image connections only from the single corresponding pixel in the input image, the dynamics of the network are given by:

$$x_i^{(t)} = \tanh(\beta[(\sum_j W_{ij} x_j^{(t-1)}) + \alpha_i I_i + b_i]) \tag{1}$$

where $W_{ij}$ are weights from other neurons, $\alpha_i$ are weights from the input image and $b_i$ is a bias term (**Fig 1E**). Since in the tag propagation solution the next time step depends only on the value of the input pixel corresponding to the $i$-th unit and the current value of the $i$-th units neighboring pixels, we can reduce the incoming weights for each neuron to five weights: four weights $W_{ij}$ for $j$ corresponding to the nearest neighbor pixels in the image and the input-image weight $\alpha_i$. Together with the bias parameter $b_i$ this yields six parameters per neuron.

Furthermore, we can take advantage of an additional symmetry. Each of the neighbors should have the same effect on a neuron's state: if any of them has a connected tag and the pixel is on, then it should also have a connected tag in the next step. Thus, the four $W_{ij}$ can be reduced to a single $w_i$ that is shared among the four pixels. Finally, as the same computation should be performed at each neuron, we can have the weights identical for all neurons, i.e., we drop the dependency on the neuron index. Taken together, the conditions for these weights to perform the tag propagation correctly are therefore given by a set of three inequalities:

$$b - 2w + \alpha > 0$$

$$b - 4w + \alpha < 0$$

$$b + 4w - \alpha < 0 \tag{2}$$

The first equation expresses the condition that a pixel should become activated if its input is on and if at least one neighboring pixels is on (i.e., the three others are off, contributing $1 \times (+1) \times w + 3 \times (-1) \times w = -2w$). The second equation expresses the condition that a neuron should be off if its input pixel is on yet all its neighboring pixels are off (contributing $4 \times (-1) \times w = -4w$). The third equation expresses the condition that a pixel should not be activated even if all its neighbors are on (contributing $4 \times (+1) \times w = 4w$) but its input pixel is off. Note that the above inequalities hold specifically for pixels not on the edges or in the corners. In these other locations, the inequalities need to be modified to account for there being only 3 and 2 neighbors respectively (or the input image can be zero-padded around the edges). These inequalities are based on a single step. If the neurons are binary, as in the original tag propagation proposal, or if $\beta$ is high, the same solution holds for the full series of time steps.

Simulating these analytically derived weights, we find the network reaches zero error if run for a sufficient number of time steps (**Fig 1F**), as expected.

## The advantage of recurrence in edge-connected pixel task stands up to a large-scale network architecture search

Having established the tag propagation solution in recurrent architectures, we systematically explored how multiple different types of architectures solve the edge-connected pixel task by training networks to output the correct edge-connected classification of training examples by stochastic gradient descent or SGD (**Fig 2A** and Methods). For each architecture, we considered networks with various numbers of layers, from only a few layers which should be easily trainable but potentially less powerful up to thirty layers. We note that–generally speaking– despite deep learning's great success, training networks by stochastic gradient descent is not guaranteed to find optimal solutions [23]. To ameliorate this issue, we first performed extensive searches on the space of learning parameters, e.g., learning rates and batch size, in a process known as hyperparameter optimization (see Methods). To further ensure finding successful models, we performed hyperparameter optimization separately for each architecture type and network depth (**Fig 2B** and Methods). Second, after we discovered the best learning parameters based on the learned network performance, we trained a large number of networks from independent initializations since starting from distinct initial conditions may lead to convergence to a different local optimum (**Fig 2B** and Methods). We chose to study the best solutions from these independently initialized networks.

We first evaluated the performance of a deep feedforward architecture network. This architecture received the image as input to the first layer and was composed of multiple hidden units in each of several layers. The output was inferred from the activation of units in the last

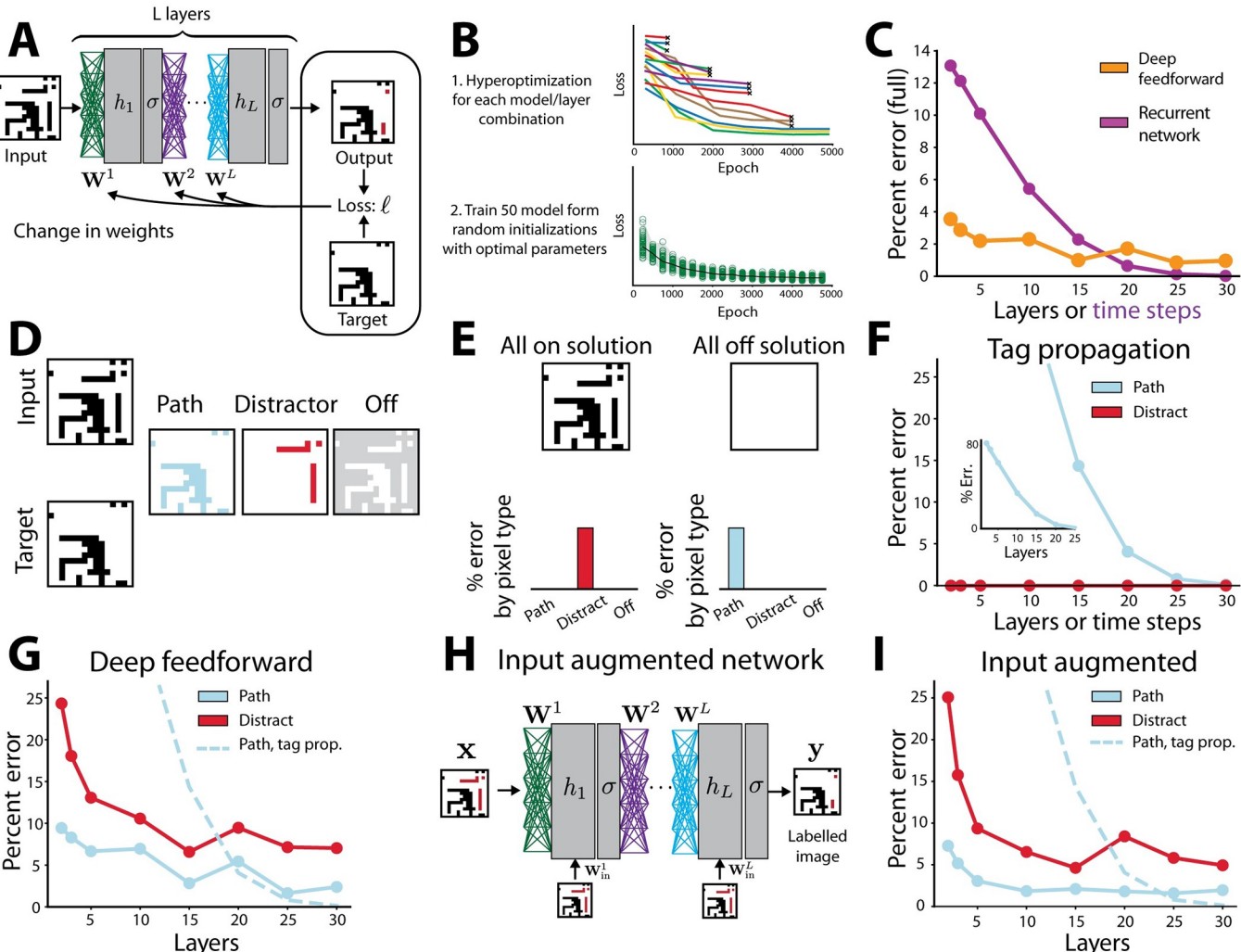

**Fig 2. Feedforward network performance on the edge-connected pixel task.** (A) Training procedure for neural networks. Weights are learned iteratively. At each iteration the weights are updated using the gradient of the loss function calculated via backpropagation. The network illustrated here is a deep feedforward network. The image is given as input to the first layer and the activation of units at the last layer is the network's output. (B) Schematic of the two-part procedure used to obtain good solutions despite the stochasticity of learning and the large parameter spaces. First, hyperoptimization is performed over the training parameters for each model and layer combination (top). Each line corresponds to a network being trained, colored differently to indicate the different hyperparameters used. The worst performing models are eliminated at regular intervals (see Methods for full details). Second (bottom), 50 models are trained from random initialization, all using the optimal parameters found in the hyperoptimization procedure. (C) Performance of the best trained solution for the deep feedforward network across layers vs. the recurrent tag propagation solution. X-axis corresponds to number of layers in the network for the feedforward solution, and number of time-steps the network is allowed to run for the recurrent solutions, which have only one layer of recurrently connected neurons. (D) Illustration of the splitting of pixels, and associated errors, into three groups: path, distractor, and off. (E) Breakdown of errors for two naïve solutions: all on, outputting all the on-pixels as connected, and all off, outputting no pixels as connected. (F-G) Decomposition of error by pixel type for each model. (F) Tag propagation implemented via recurrent network. X-axis corresponds to the number of time-steps the network is allowed to run. Blue dots and line correspond to errors on path pixels, red dots and line correspond to errors on distractor pixels. Inset shows same data with larger axis range. (G). X-axis corresponds to number of layers in the network. Blue dots and line correspond to errors on path pixels, red dots and line correspond to errors on distractor pixels. Dashed line corresponds to tag propagation error on path pixels for reference. (H) Schematic of input augmented architectures where the full image is added as input at each layer, allowing the tag propagation solution to be part of parameter space. (I) Decomposition of error by pixel type for the input augmented network. Same plotting convention as G.

layer, referred to as the output layer (**Fig 2A**). Its input-output function is given by:

$$\mathbf{y} = \sigma(\mathbf{W}^L(\dots\sigma(\mathbf{W}^1\mathbf{x} + \mathbf{b}^1)\dots) + \mathbf{b}^L) \qquad (3)$$

Here, σ is the tanh function, $\mathbf{x}$ is the input image, the matrix $\mathbf{W}^1$ is the weights from the input to the first layer, $\mathbf{W}^L$ the weights from one layer to the next and $\mathbf{b}^L$ is a bias term. We note that in following equations we will drop the explicit notation of the constant bias term to simplify presentation, absorbing it in the standard way into the weights by adding a fixed input to all training examples ([24]; see Methods). We set the size of the hidden layers to be the size of the input $\mathbf{x}$. Thus, the network had $L(N^2 \times N^2)$ parameters for the fully connected linear weights $\{\mathbf{W}^1,...,\mathbf{W}^L\}$ and $LN^2$ parameters for the biases $\{\mathbf{b}^1,...,\mathbf{b}^L\}$ for a total of $L(N^4+N^2)$ parameters. This network architecture is illustrated in **Fig 2A**. Note that due to the common problem of vanishing gradients in deep feedforward models, a small number of skip connections were added from the input to deeper layers for models with 15–30 layers, which we found improved network performance (see Methods).

We began by measuring the raw accuracy of trained networks. For each layer number we trained 50 networks from distinct random initializations using layer-number specific hyperparameters for 5000 epochs and displayed the best network's performance on a held-out dataset (**Fig 2C**). We evaluated the error of each network by outputting an $N \times N$ image that predicts for each pixel whether it should be labelled as connected. Error decreased with increased layer number, indicating we successfully trained networks even when they had many layers. Moreover, we found that deep feedforward networks were able to find approximately accurate solutions even with hidden layers of a size no larger than the number of pixels, a solution far more efficient than the exponential scaling of the straightforward solution of exhaustive enumeration in a single hidden layer previously suggested (**Fig 2C**).

To better understand the solutions found by the deep neural network, we compared the observed errors to those of the tag-propagation solution. The tag-propagation solution, if allowed to run till convergence, will yield zero error on all types of pixels. However, if stopped after a fixed number of iterations it will only correctly label connected pixels close to the edge, yielding non-zero error (**Fig 2C**). We found that if time is a significant limitation, meaning that network architectures can only propagate for a few time steps or through a small number of layers, then the feedforward solutions learns far more accurate solutions (**Fig 2C**). With just two layers, the deep feedforward network was able to learn solutions with 3.54% error vs. 13.1% error in the tag propagation solution.

Before analyzing the error in more detail, however, we noted that accuracy on all pixels is not the most informative measure because it conflates multiple types of error. For instance, pixels that are off in the input will never be connected and labeling all pixels that were off as unconnected is an easy way to reduce error on off-pixels to zero. Therefore, to obtain more informative metrics, we subdivided the error by mutually exclusive pixel types (**Fig 2D**). The first group of pixels are "path" pixels which are on and connected to the edge by other on-pixels. An error on these pixels indicates a failure to identify all connected pixels. The next were "distractors": on pixels disconnected from the edge, which should be labelled as disconnected, and "off" pixels which are off in the image and should all be labelled as disconnected. As an intuition for this breakdown of pixels and error type, a network that simply labeled all pixels as connected if their value in the input image was on and disconnected if their value in the input image was off would score zero error in path pixels, zero error in the non-activated pixels but a full error (i.e., 100 percent error) in the distractor pixels (**Fig 2E**). Conversely, a network that labels all pixels as disconnected would yield zero error in distractor and disconnected pixels and a full error in the path pixels (**Fig 2E**).

The relative number of pixels in these categories depended on the way input images were generated. Intuitively, the two challenging aspects of the task are long paths and the presence of distractors. As expected, we found that the two most important properties that increase task difficulty were the presence of extended-length paths connected to the edge of the image and

the percentage of distractor pixels. Images were generated by seeding pixels randomly and continuing a path randomly based on these seeds (see Methods). Using this sampler generation method, images had on average 15.9% of their pixels that were on and connected to the edge and 8.3% of the pixels that were on and disconnected from the edge (i.e., distractors) with the remaining 75.8% pixels off. Other choices of sample generation yielded different ratios of pixel types and qualitatively similar results across architectures as long as they contained a substantial number of challenging examples.

By design, the tag propagation networks were guaranteed to make no errors in non-activated pixels or distractor pixels (**Fig 2F**). We found that deep networks learn different forms of solutions, producing a mixture of errors on path and distractor pixels (**Fig 2G**). This was still true even for deep learning networks with twenty or more layers, by which point the tag propagation solutions became more effective.

The feedforward networks as defined above are the most natural extension of previously studied single hidden layer networks and the standard architecture of fully connected networks in deep learning. As formulated, however, there was an important mismatch between them and the tag propagation architecture. Namely, although the input can filter through the layers, it only directly affects the bottom layer; in recurrent networks (and the tag propagation solution), the network interacts with the input at each time step. To address this issue, we trained an additional network architecture, which we refer to as input augmented, where each layer also receives the original image as input (**Fig 2H**). The input-output function for such architectures is given by:

$$\mathbf{y} = \sigma(\mathbf{W}^L(\ldots \sigma(\mathbf{W}^1\mathbf{x} + \mathbf{W}_{\text{in}}^1\mathbf{I})\ldots) + \mathbf{W}_{\text{in}}^L\mathbf{I}) \tag{4}$$

where $\mathbf{W}_{\text{in}}^k$ are the weights from the input to the hidden units of the kth layer, each of which is $N^2 \times N^2$. Thus, the total number of parameters in the network is $L(N^4 + N^4 + N^2) = L(2N^4 + N^2)$ parameters.

The input augmented networks slightly outperformed the standard feedforward networks both in terms of overall performance and the number of layers needed to achieve it (**Fig 2I**). The best overall performance by the deep feedforward network was 1.63% error on path pixels and 7.14% error on distractor pixels (an average of 3.52% error, not counting off pixels) at 25 layers. Meanwhile, the input augmented network achieved its best performance with 2.10% error on path pixels and 4.63% error on distractor pixels (an average of 2.97% error not counting off pixels) at 15 layers. Both architectures still underperformed relative to the recurrent network solutions, even though the tag propagation solution was in their parameter space. In other words, we know that a solution with close to 0% error could have been learned by the input augmented network, but the error breakdown showed that this solution was not learned.

In summary, even with the computational resources of modern deep learning, the purely feedforward networks we trained did not achieve the same task performance as the simple recurrent tag propagation network. On the other hand, we found that feedforward solutions did not converge on naïve poor-performing solutions, such as attempting to enumerate an exponential number of solutions. Instead, they were able to learn compromise solutions that were more effective than tag-propagating in the low layer or timestep regime, demonstrating the importance of empirically studying solutions found by deep learning.

## Partial architectural constraints enable learning of highly effective tag-propagation-like solutions

To better understand why the tag-propagation solution was not learned despite being an achievable solution of the input augmented networks, we tested three new network

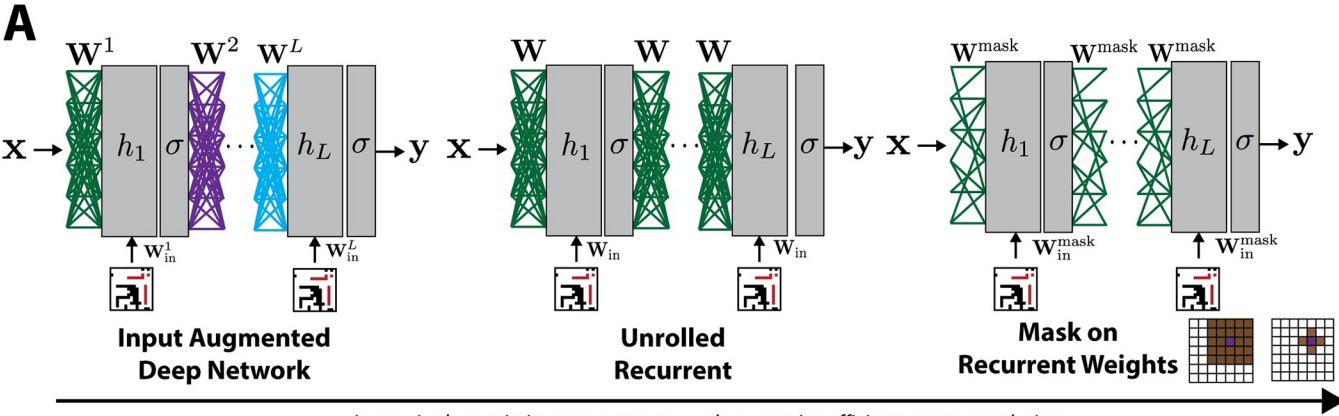

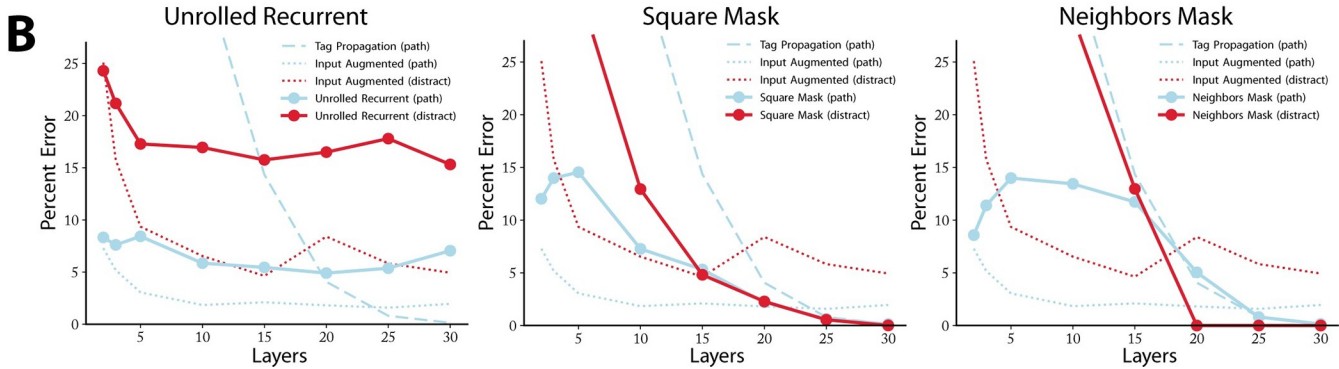

**Fig 3. Masked recurrent networks performance on the edge-connected pixel task.** (A) Schematic of the addition of constraints on the feedforward parameter space generating an increasingly restrictive parameter space that still contains the efficient tag-propagation solution. From left to right: weight sharing across the layers creates an unrolled recurrent network, masking of shared weights (i.e., enforcing locality in the operations) to a grid around each pixel creating unrolled recurrent networks with sparse weights (middle), masking to just the nearest neighbor of each pixel (right). (B) Decomposition of error by pixel type for each model. X-axis corresponds to number of layers in the network. Blue dots and line correspond to errors on path pixels, red dots and line correspond to errors on distractor pixels. Dashed line corresponds to tag propagation error on path pixels for reference. Dotted blue and red lines indicate input augmented architecture error on path (blue) and distractor pixels (red). Each subpanel corresponds to a different network architecture. From left to right: unrolled recurrent, square mask, neighbors only mask.

architectures that are gradated reductions of the solution space. These still contain the tag-propagation solution, but with an order of magnitude fewer synapses (**Fig 3A**). Each reduction was motivated by an invariance or a locality in the tag-propagated solution.

The first invariance we considered in the tag-propagated solution was the invariance across time. Each time step performs the same calculation, and thus we first consider a network with weights shared across layers (i.e., limiting the network to only one matrix **W** between hidden layers, one matrix of input weights $\mathbf{W}_{\mathrm{in}}$, and a single vector of biases **b**). The input-output function of this architecture is given by:

$$\mathbf{y} = \sigma(\mathbf{W}(\ldots\sigma(\mathbf{W}\mathbf{x} + \mathbf{W}_{\mathrm{in}}\mathbf{I})\ldots) + \mathbf{W}_{\mathrm{in}}\mathbf{I}) \tag{5}$$

As before, **b** has been absorbed into the weights. Such a solution is sometimes referred to as a "rolled-out" recurrent network, since the input-output function is identical to a recurrent network running for the number of time steps equal to the number of layers ($L$) driven by a constant (in time) input **I**. This invariance causes the number of parameters to be substantially smaller and not to scale with the depth of the network; the total number of parameters in the network is $N^4 + N^4 + N^2 = 2N^4 + N^2$.

Like the input augmented network, the trained unrolled recurrent network achieved much better performance than the tag propagation solution at low layer numbers, but it quickly plateaued in performance and was surpassed by the tag propagation solution at 15 layers (**Fig 3B**). The breakdown of error revealed that the network learned a compromise solution with 17% to 25% error on the distractors, but the performance was noticeably worse across all layers than the input augmented solution. The most likely cause of the increased error was that many solutions learned by the input augmented network were eliminated through the restriction to shared weights across layers, i.e., the reduction of the parameter space. However, this restriction still did not enable the learning of a successful tag propagation-like solution. This provides further, though indirect, evidence that the input augmented architecture learned solutions different from approximate tag-propagation.

The next two architectures imposed further constraints based on the locality of the tag-propagation solution. In the analytical solution (**Fig 1F**), each neuron performed its calculation based only on its four immediate neighbors, and thus the weight matrix was highly sparse with a pattern based on locality, i.e., which pixels are close to each other in the image. This motivated us to study masking the weight matrix to induce locality-based sparsity (**Fig 3A**). We denoted the masked weights as $\mathbf{W}^{\text{mask}}$ and $\mathbf{W}_{\text{in}}^{\text{diag}}$, as the recurrent weights will have a sparsity pattern determined by the structure of the image and the input weights will be masked so that each neuron only gets input from its associated pixel value in the input.

We considered two levels of locality restrictions. In the first, we used an $r \times r$ square mask centered on the pixel associated with the hidden units and one weight from the corresponding pixel in the input. Each mask had $r^2$ parameters and therefore $\mathbf{W}^{\text{mask}}$ has $r^2 N^2$ parameters while $\mathbf{W}_{\text{in}}^{\text{diag}}$ just has $N^2$ parameters, one for each pixel, for a total of $r^2 N^2 + N^2 + N^2 = (2+r^2)N^2$ parameters. We used $r = 5$, therefore the network had $27N^2$ parameters.

Second, we considered a network with a mask that enforces the exact locality structure of the original task such that $\mathbf{W}^{\text{mask}}$ had $4N^2$ parameters corresponding to one weight for each pixel's four nearest neighbors. Each hidden unit also had one weight tying it to the relevant input pixel in $\mathbf{W}_{\text{in}}^{\text{diag}}$ and a bias. Therefore, the total number of parameters is $4N^2 + N^2 + N^2 = 6N^2$.

We found that at high layer number (25 and 30 layers) the masked networks learned a solution that was as effective as the tag propagation network (**Fig 3B**). Interestingly, at low layer number (e.g., 10 and 15 layers) the masked networks did not just learn the tag-propagation solution which has lower performance at this number of layers, but rather learned a solution with an error profile similar to successful solutions found at low layer number by the input augmented networks (**Fig 3B**). Thus, the masked networks as a class learned an effective interpolation between a compromise solution at low layer number and the tag propagation-like solution at high layer number (**Fig 3B**). The switch between the solution types occurred around 20 layers when the tag propagation solution becomes more advantageous, and the masked networks learned this solution instead. The 5×5 masked networks were more successful at this interpolation: at low layer numbers, they used the additional weights to learn more complicated and effective solutions, but the enforced locality structure was sufficient for learning to discover a tag propagation-like solution at high layer number. Thus, the masked architectural constraints that pushed the network towards more local solutions were not so constrictive as to prevent any other form of solution. In other words, this constraint enabled networks to learn effective solutions across a range of time scales.

**Fig 4A** summarizes the key difference between the trained feedforward and recurrent networks on the edge-connected pixel task. At low layer number or time step, the feedforward outperforms the recurrent architecture with a similar number of neurons, even when

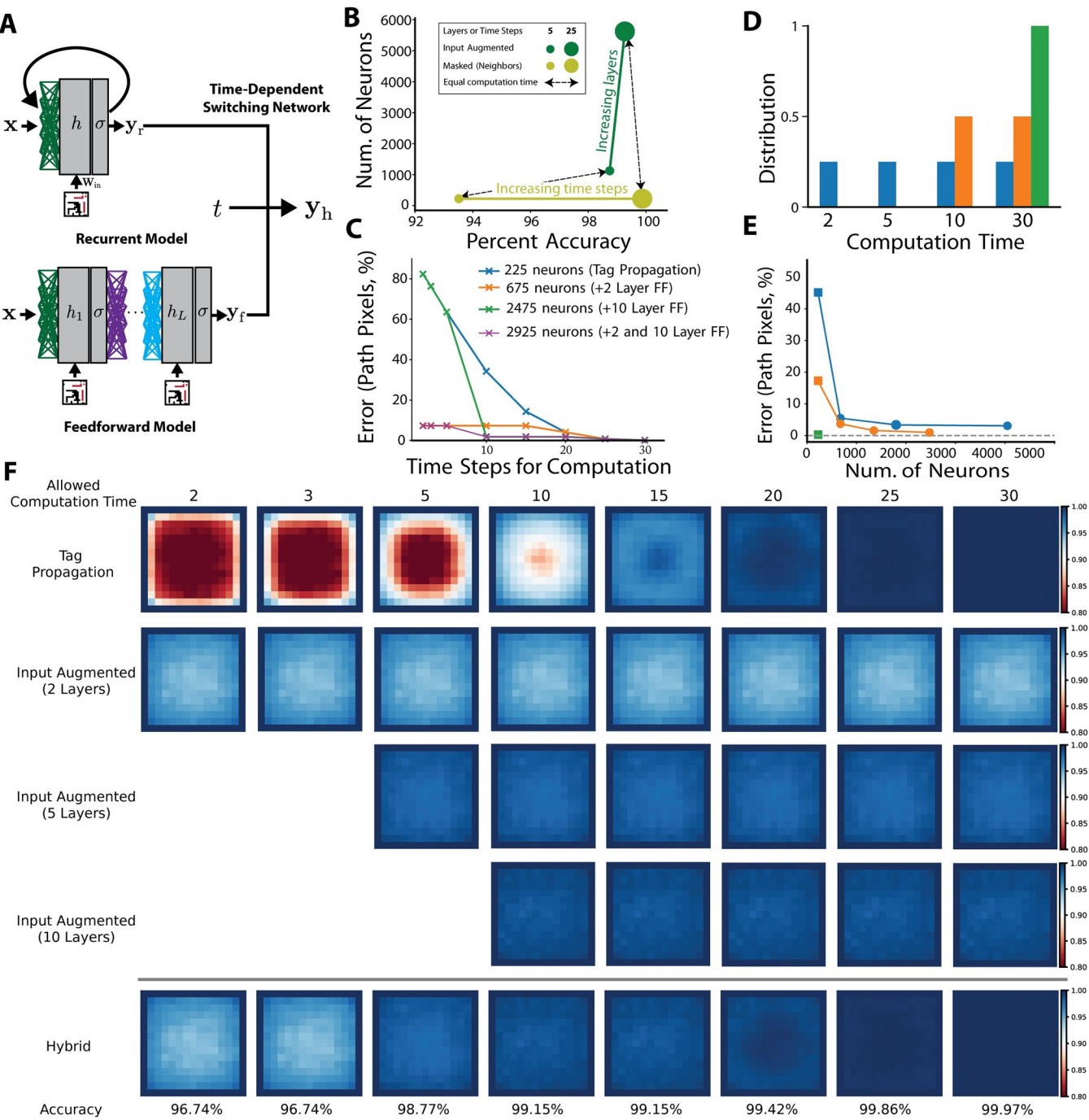

**Fig 4. Performance comparison of uniform and hybrid networks on the edge-connected pixel task.** (A) Schematic of a general hybrid network combining a recurrent and feedforward model. For any time-step $t$, the switching network combines the current output of the recurrent network $y_r$ and the output of the feedforward network $y_f$ to form the hybrid output $y_h$. Note that the feedforward output can only be used by the switching network when $t$ is greater than or equal to the number of layers in the model. (B) Performance of the best solution after hyperoptimization type plotted against the number of neurons to motivate why combining these two architectures into a hybrid networks can provide better performance across a range of computational times. The feedforward architecture (input augmented) provides a superior neuron/performance tradeoff at low computational time while the recurrent architecture (masked neighbors) provides a superior neuron/performance tradeoff at high computational time. Colors correspond to different network architectures. Small circles correspond to recurrent networks run for five timesteps and feedforward networks with five layers. Large circles correspond to architectures that use 25 timesteps or layers. Solid colored lines connect models of the same architecture with different layers or time steps. Dotted lines with arrows highlight the masked and feedforward architectures with equivalent layers or timesteps for ease of comparison. (C) Performance of hybrid architecture designed to be able to switch on-the-fly based on how many time steps are available to output a labelling. Here we consider the simplest switching network which can choose to either to output $y_r$ or $y_h$ at a given time step. The model combines the tag propagation solution and trained input augmented networks; the budget for each curve is

the number of neurons the full model requires. The blue curve is the error profile for tag propagation alone while the orange curve shows the result of combining tag propagation with the 2-layer feedforward (abbreviated FF) model. The green curve shows tag propagation combined with the 10-layer feedforward network and the purple curve shows the combination with both the 2-layer and 10-layer networks. Note that the figure only shows path error; the feedforward solutions will also have some error on the distractor pixels. (D) Three distributions over computation time: the blue distribution is evenly split over short, medium, and long computation times (2, 5, 10, and 30 steps) and the orange distribution is evenly split over medium and long computation times (10 and 30 steps). (E) Performance of hybrid networks on three distributions of allowed computation time illustrated in panel D, meaning the fraction of runs the network is limited to a certain number of steps. The x-axis corresponds to the number of neurons allowed when constructing the hybrid solution. The first point on all curves allows only a fully recurrent architecture and thus is not hybrid. This is indicated by a square marker. The rest of the x-axis corresponds to a neuron budget allowing hybrid solutions with increased feed-forward composition. Additional hybrid models were only included in the plots if they improved the performance over models with fewer neurons (see Methods). (F) Per-pixel switching hybrid network learned from observing the output of the four networks (tag propagation; input augmented with 2, 5, and 10 layers) on 10,000 test samples. The heatmap shows the percent of times a given network correctly classified a pixel.

considering the effective solution learned by the masked network. But at high layer number or time step, the recurrent network learns a near-perfect solution without an increase in neurons while the feedforward network only slightly improves performance at a large cost of additional neurons. This shows that the two network types excel in different computational regimes, the feedforward network at low computation time and the recurrent network at high computation time. We now consider how these two architectures can be combined into a hybrid architecture that performs well in both regimes.

## Hybrid architectures are effective across timescales, and outperform pure recurrent architectures when computation time is limited

Hybrid architectures that combine recurrent and feedforward architectures can inherit the best aspects of each architecture. We therefore determined whether models that incorporated both recurrent and feedforward architectures combined through a mechanism for switching the readout from feedforward to recurrent after a given number of timesteps might be computationally most robust (**Fig 4A**). Such architectures utilized the early approximate solutions of feedforward networks to improve early prediction (green points in **Fig 4B**) and the ability of recurrent networks to generate accurate solutions at longer timesteps (yellow points in **Fig 4B**). For example, if we compare tag propagation with the trained 2-layer input augmented network, we see that the feedforward model outperforms tag propagation until 20 time steps. This is because it takes many time steps to propagate the tag along long paths. If we initially set the solution to that provided by the feedforward and only switch to the recurrent network at 20 time steps (if an answer was not required sooner), we achieve a much better error profile across a range to time scales for a small increase in neuron budget, e.g. 7.28% error on path pixels after 5 timesteps vs. 63.52% error from tag propagation alone (orange curve in **Fig 4C**). For increasing allowed number of neurons, hybrid architectures offered increasingly better solutions (**Fig 4C**). Given a particular distribution of constraints on allowed computation timesteps (e.g., those shown in **Fig 4D**) one can derive the best choice amongst these hybrid architectures for a given neuron budget (**Fig 4E**). In fact, performance was substantially improved by simply adding a 2-layer feedforward network to a recurrent solution (**Fig 4C** compare orange to blue line, **Fig 4E** second point in curves). Thus, networks designed to have hybrid architectures can display greatly improved performance across a wide range of constraints on computation time. We note that this hybrid architecture is different from the model considered in [21], in which a confidence-based threshold was used to make a flexible trade-off between accuracy and computation in a purely recurrent model. The model in [21] would more closely correspond to varying the number of time steps used for tag propagation.

We next studied networks that learned to perform more granular, per-pixel switching. The recurrent network used was the tag propagation network. The per-pixel accuracy calculated on

10,000 test samples (generated in the same manner as those for training) improved over time as described before, with a clear spatial pattern consistent with propagation from the edges as expected (**Fig 4F** top). Three feedforward input augmented networks were then trained on the task with 2, 5, and 10 layers. To make the relation to computation time comparable across feedforward and recurrent architectures, we adopted the convention of the feedforward network requiring one time step per layer and no additional computation at timesteps greater than the number of layers. Thus, if the computation time is less than the number of layers in the network, the output does not reflect the current input (which we considered as lack of an available solution rather than generating random output). Feedforward solutions were more uniform in their error across the image and (as noted before) performance improved with higher layer number (**Fig 4F** middle). For a given allowable computation time, the switching network learned a mask that selected one of the available network outputs for each pixel based on the straightforward rule that selects the network with the highest accuracy for that pixel as calculated on the 10,000 task samples. As expected, the switching network yielded high performance across a range of allowable computation times and across the image, again demonstrating the utility of combining recurrent and feedforward architectures.

## Relation to convolutional architectures

Here we place the architectures considered for the edge-connected pixel task into the context of convolutional architectures. Instead of implementing tag propagation in a recurrent architecture, each individual step of the tag propagation solution can be implemented via a 2D convolutional layer followed by a hyperbolic tangent non-linear activation function. More specifically, to re-express the tag-propagation solution, the convolutional layer should be composed of two input and two output channels; the first output channel is used to perform propagation (which requires the previous step and the original image as input) and the second output channel is used simply to simply copy the input image (for more details see Methods). If at each step the input is zero padded and the original input is the edge pixels that are on, repeated application of this block will perform the tag propagation algorithm. This will be a highly inefficient solution in terms of number of neurons if implemented as a feedforward convolutional architecture as it performs the same computation with number of neurons equal to the image size times number of layers. However, due to the weight structure, the solution can be equivalently implemented in a recurrent convolutional architecture. The intermediate masked networks we trained can also be seen as single channel recurrent convolutional layers with local connectivity, i.e., without spatial weight sharing such that each pixel learns its own filter.

Lastly, as the edge-connected pixel task is in essence a binary segmentation task, we compared performance to an architecture explicitly designed for image segmentation, U-Net architectures [25]. In brief they are comprised of a contractive path (from large filters to small filters) followed by upsampling to return to the same image size and assign an output to each pixel in the image. Two important features of the architecture are that in the contractive path the number of channels is increased (mirrored by a decrease in the upsampling part) and that the final output of each block of the contractive path is also concatenated to the upsampled features yielding multi-scale information (see Methods for full architecture). We found that on the one hand, despite these networks' known strength in segmentation, when using a similar number of neurons as the input augmented network, the U-Net was not able to learn the edge-connected pixel task. On the other hand, if the number of neurons was greatly increased by doubling the number of channels in each layer resulting in 126 times the neurons in the tag propagation solution and 5 times the neurons in the 25-layer input augmented network, the

U-Net was able to achieve performance similar to the tag propagation algorithm and the masked networks. This most likely indicates that introducing the inductive bias of the architecture aids in learning the task in large networks just as it did in other segmentation tasks. However, the recurrent networks we consider remain much more efficient in the number of neurons even compared to this class of networks specifically used for segmentation.

### Generalized tag propagation and temporal connectedness: Application to decision-making tasks

Having used the edge-connected pixel task to analyze the differences between feedforward and recurrent computation, we next sought to generalize the understanding of regimes in which recurrent computation is particularly effective by introducing more abstract notions of connectivity and their implementation using an architecture inspired by tag propagation which we term generalized tag propagation networks. Specifically, the notion of connectedness does not have to be limited to the spatial domain comprising connectedness of pixels and objects, but can be extended to more abstract domains, and in particular, the temporal domain. Indeed, all animals need to use current information to judge the possible outcomes of different actions, yet the temporally-extended nature of most actions leads to the success of an action depending not just on the current state of the world, but rather also on its future state. As not all future states are connected to a given current state, the computation of which future states are possible can be seen as analogous to asking which spatial locations are connected to a given spatial location. As such, computations that make predictions about future states of the environment are necessarily functions of global input, raising the possibility that extensions of the tag propagation framework may be useful for their implementation.

Before developing a generalized tag propagation framework for application to predictions about future states of the environment, we first detail the different parts of the tag propagation process used in the edge-connected task above. To calculate the state of the edge-connected tag at each pixel, we first set the tag to positive at a small set of (seed) pixels where local information was sufficient (i.e., the edge pixel). We then propagated the tag sequentially to neighboring pixels along a given connectivity structure (a two-dimensional grid where each pixel is connected to its four neighbors). At each downstream pixel, we determined whether the tag should be propagated to that pixel according to a straightforward propagation rule: if the input to a pixel is positive and one of its neighbors' tag is positive, the tag of that pixel is set to positive. Finally, once propagation completed, we computed the output at each pixel by the most straightforward local function of the input to a pixel and its tag: if the tag is positive the output should be positive. Each of these core components of the tag propagation framework–(i) number of tags, (ii) determination of propagation seeds, (iii) the adjacency structure, (iv) propagation rule, and (v) post-propagation output calculation–can be extended to fit complex tasks, prompting us next to evaluate which generalizations would accommodate the structure of the temporal prediction problem.

Why might the future prediction of environmental states be challenging for the basic tag propagation framework? The most crucial issue is that now the environmental states are not frozen, nor do they depend only on the agent's own actions, but rather can change independently of the agent. Consider, for concreteness, an example computation a foraging animal may need to implement when they notice a new piece of food, i.e., to decide whether to pursue it or leave it knowing that another competing animal is more likely to get there first. The success of the attempt to collect the food clearly depends on the future actions of other animals (and other properties of the environment). This problem is both non-local spatially, e.g., dependent on the positions of competitors and obstacles, as well as non-local temporally, e.g.,

dependent on the future actions of competitors. Yet despite its complexity, it can be conceptually captured within the generalized tag propagation framework by deriving the values of future states through the outcome and interaction of propagation of multiple tags as we describe below.

We formalized this task as follows: four $N \times N$ binary inputs were given to a network, each of which was designed to encode information about a spatial environment. Input one was a random environment consisting of several barriers represented by the value -1. The size of the environment was always the same, and no animal could move onto or through squares that were included in the barrier. Input two was the location of the animal in this environment, input three was the location of an arbitrary number of competitors, and input four was the location of the food all animals were trying to obtain (**Fig 5A**). The task for the network was to appropriately output a "run" or "stay" signal based on whether or not a direct move to the food by the animal will be successful. We note that this task does not of course capture all the complexities possible in judging the future utility of actions, e.g., it assumes the animal has perfect knowledge of barrier and competitor locations which would require some exploration in most cases. Rather it is meant to demonstrate key features of such problems and why generalized tag propagation can be an effective solution. Indeed, from an algorithmic perspective, the task can be reduced to determining the relative distance between sets of nodes in a non-uniform connectivity space (see Methods).

Just as the edge-connected task could be solved efficiently by a network that propagates a tag, the competitive foraging task can be solved efficiently by a network that propagates multiple, distinct tags. We refer to this solution as generalized tag propagation. Namely, two distinct tags are propagated by two recurrent networks; one network propagates a tag based on the animal's location, and another network propagates a tag based on the competitors' locations (**Fig 5B**). These networks' output is then combined with an input describing the obstacles and food location to an intermediate representation from which we generated the architecture's full output, a binary decision to pursue the food or not (**Fig 5C**). Similar to the edge-connected task, solutions for the weights of the propagation networks can be derived analytically and implemented in a recurrent network that then outputs the correct decision labels (see Methods). In essence, this solution learns to generate a tag starting at the location of the animals (self or competitors), propagating out each time to any neighboring pixel that is not a barrier in the environment. In this manner, the hidden state at each time represented all points reachable by the animal or group of competitors at time point $t$. Note that this tag does not represent a specific path taken by the animal, but rather the range of all possible points a path might reach. Given these tags, the output depends on reducing non-local information to a local computation on the state of the tags at the food location. We refer to this solution as the generalized tag propagation algorithm.

For simplicity, we trained network architectures separately on the two parts of the proposed solution: (i) generating the two tags and (ii) learning the output from the intermediate representation (tags and food location) to the decision. Trained networks were able to successfully perform both parts of the task. First, trained recurrent neural networks accurately learned the propagation (**Fig 5D**). The propagation networks received input describing the barriers and the initial location of the animal and competitors. With these inputs, each network was trained by stochastic gradient descent to output the correct tag for each pixel at a given number of timesteps. In essence, the network should return all locations accessible to the animals in the specified number of time steps (see Methods for training details).

Similar to the edge-connected task, we trained the network to produce the correct labeling. Here, though, the state of the propagation at specific times, which reflects the possible paths up until that time point, were of interest (rather than just the last propagated step as in the edge-

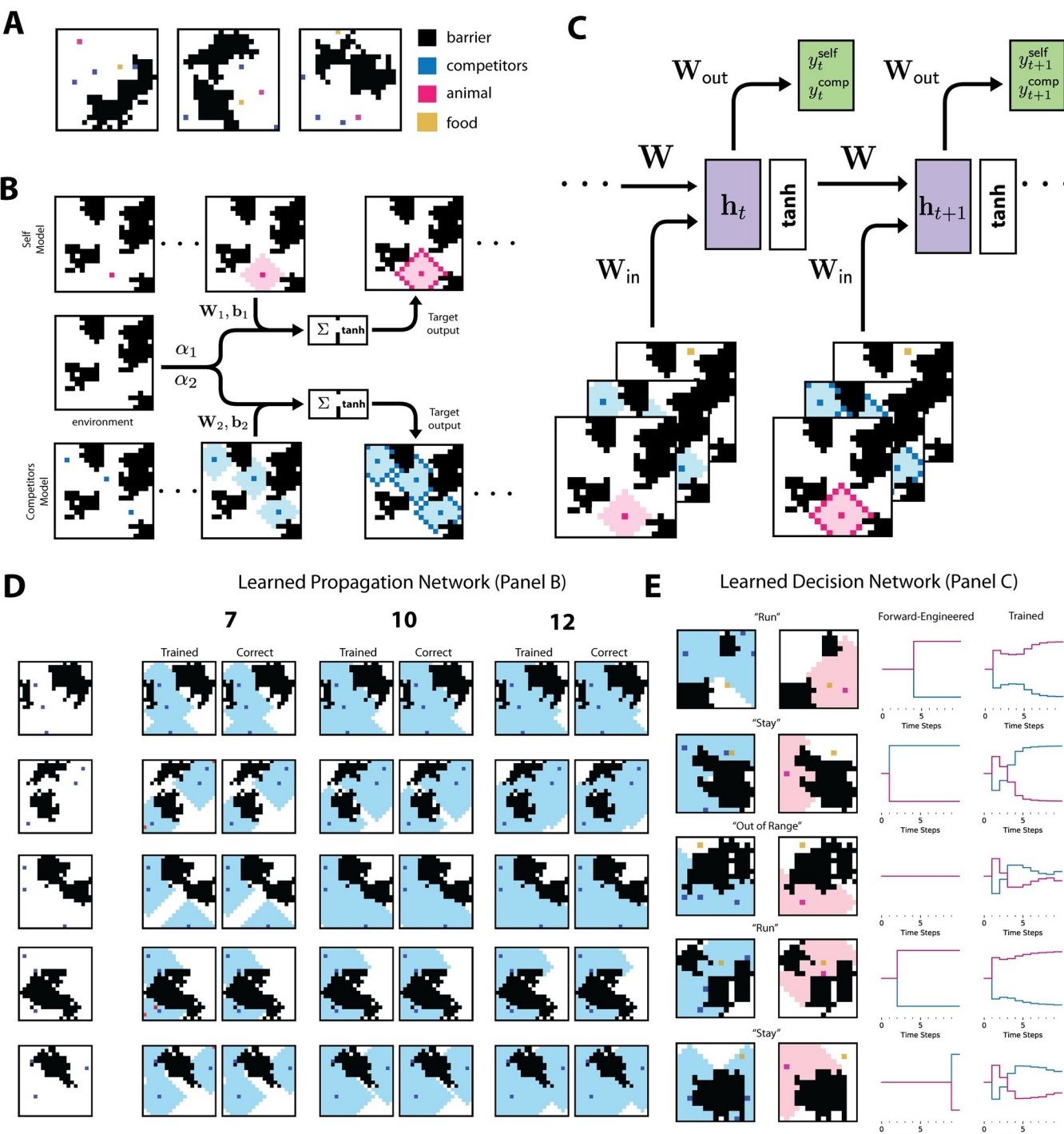

**Fig 5. Competitive Foraging Task.** (A) Randomly generated examples of the competitive foraging task. (B) The animal's and competitors' propagation networks. Each one implements the tag-propagation algorithm with open space pixel corresponding to on pixels and barriers corresponding to off pixels. Unlike the edge-connected pixel task, the source pixels change in every sample to correspond to the location of the animal and its competitors respectively. See Methods for further implementation details. (C) Architecture for the trained decision network. The time series of animal and competitor ranges are concatenated with the food pixel and then are used as the input to a recurrent network. The decision traces shown in panel E are the projection of the trained network onto the two-dimensional readout. (D) Sample outputs for the trained propagation networks. These are the best networks trained with 7, 10, and 12 layers respectively. Errors in the trained network are marked in red. We show results only for the competitors' network as the propagation task is the same for both the animal and its competitors; only the input which corresponds to the initial location changes. (E) The generalized tag propagation implements a correct version of the decision trace and is described in the Methods. For comparison, we show the decision trace outputted by the trained decision network. Propagation is shown after ten time steps, and each example is labelled with the correct decision after ten steps: "stay," "run," or "out of range." Note that if the food location is in the range of both groups of animals, the decision is based on which animal can reach the food first.

connected task). We trained RNNs to run for 7, 10 and 12 timesteps to produce the correct labeling following that number of timesteps (see Methods). Networks successfully learned the propagation: the best 7-layer network achieved 3.50% error; the best 10-layer, 0.0015% error; and the best 12-layer, 0.0009% error (**Fig 5D**). Trained networks generalized well across number of time steps used: the best 10-layer network achieved 0.0004% error on the 7-step propagation task and 0.0015% error on the 12-step propagation network.

Notably, the trained propagation demonstrates another benefit of the tag propagation algorithm. Learning the competitive foraging task requires generalizing the propagation rule to arbitrary starting points. Stated in terms of the edge-connected pixel task, the equivalent of edge pixels which serve as the "source" or seeds of propagation are now wherever the animals are initially located, and this changes from example to example. The tag propagation algorithm has this generalization built in since it gives a correct update rule for any current state of the tags.

The second part of the calculation, distilling the tags and food location into a decision, is conceptually straightforward: if the tag that corresponds to the animal reaches the food location first then there is no path the competitors can take to arrive there first. We trained the recurrent decision network to output the correct choice based on the label propagation given by the solution to the animal's and competitors' tag propagation presented above. Specifically, we used the parameters for the best trained 12-layer propagation network as input to the decision network. The propagated tags were concatenated with the food location at each time step and then used as the input to a recurrent network with a 25-dimensional hidden state). Training the network consisted of training the input weights, $\mathbf{W}_{in}$, the recurrent weights, $\mathbf{W}$, and a set of read-out weights, $\mathbf{W}_{out}$, which map the hidden state to a two-dimensional vector (see Methods). The loss function for training this network was the binary cross entropy loss between the two-dimensional output vector after 15 steps and a two-dimensional binary label with 1 in the position of the correct decision.

The trained networks successfully learned the task, with the best trained network achieving 90.2% accuracy. To understand the nature of the solution learned by the recurrent neural network, we compared its output over time to the output over time of our analytical solution, the generalized task propagation solution (**Fig 5E**). As expected, the solutions were not identical, yet comparing the two sets of decision traces showed how the trained network achieved a similar solution by learning a more complex integration of evidence towards the decision compared to the tag-propagated time series over time. Combined, these results demonstrate that extending the tag propagation concept to multiple tags permits the efficiency with which recurrent networks can turn a global computation into a local one to be applied to problems of decision making involving temporal prediction in complex settings.

Lastly, we trained two convolutional architectures on the end-to-end binary decision tasks. The four $N \times N$ inputs were stacked as four channels of an image with the binary values represented by 0 and 255 rather than -1 and +1. We trained a straightforward CNN with 28k parameters (see Methods for architectures) and a ResNet-20 with 269k parameters [26]. Out of 10 models trained, the best straightforward CNN achieved 87.04% accuracy and the best ResNet-20 achieved 93.86% accuracy. The straightforward CNN was less accurate than both the tag propagation solution and the trained recurrent solution and was less efficient than both. ResNet-20 accuracy was relatively high, higher than the trained recurrent network, but still lower than the tag propagation solution which is able to perfectly perform the task, and the architecture was far less efficient in terms of number of neurons and parameters. For a 15×15 environment, the propagation component of the tag propagation network had 1350 trainable parameters, and the decision component had 4 parameters for a total of 1354 parameters. The trained recurrent decision network had 17.5k parameters in addition to the 1350 parameters of the propagation network for a total of 18.9k.

## Discussion

Despite tremendous progress in the design of new artificial neural network architectures, a thorough understanding of the computational advantages of incorporating recurrent connections remains an important open challenge. By performing a large-scale performance evaluation on the edge-connected pixel task for a variety of network architectures, we demonstrated that recurrent architectures implementing the tag propagation algorithm proposed by Roelfsema [8, 9] continue to outperform state-of-the-art feedforward networks when run for a sufficient number of time steps. Furthermore, a hybrid architecture which switches between feedforward architectures when only given a few time steps and recurrent tag propagation when given sufficient time steps, provides superior performance when computational time can be limited and only known on-the-fly. We then described how many decision-making tasks where an action depends on a future state can be interpreted as dependent on the connectivity between present states and future possible states. By generalizing tag propagation to multiple tags and to complex propagation algorithms, we demonstrated how more complex recurrent networks can solve at least simple cases of such tasks. While the tasks we studied were highly abstracted, our findings shed light on the set of computational tasks that can be solved efficiently by recurrent computation, how these solutions may appear in neural activity, and how they may relate to functions important in animal behavior.

What is behind the propensity of recurrent architectures to find efficient solutions in such complex, globally connected environments? One possibility is that in essence, generalized tag propagation executes a form of hierarchical processing. Indeed, in the multi-tag propagation approach for the competitive foraging task, the tag propagation layers can be thought of as introducing intermediate variables that modulate how inputs, or shallow layers, are used by the decision, or output layer. Previous work has highlighted the efficiency of hierarchical frameworks in dealing with complex inputs, including Linsley et al. [15] and Brosch, Neumann, and Roelfsema in the context of contour tracing [12] and Jehee, Lamme, and Roelfsema in the context of boundary assignment [11]. In these papers, deeper layers contain representations of coarser features that develop dependent on higher-resolution features in the layers below but also influence lower layer dynamics. In this manner, the network can iteratively extract large-scale information about the image to make global decisions about whether points are connected. Along these lines, Nayebi et al. [27] found that artificial neural networks with local and long-range recurrent feedback obtain improved performance on image classification for decreased parameter cost. In other words, intermediate computations communicate information about larger-scale features to lower levels, perhaps to shift computation to parts of the image more relevant for computation. Beyond the general use of hierarchy in machine vision to deal with complexity of inputs [28], "attention"-based networks (such as transformers for NLP [7] and graph attention networks for graph-structured data [29]), which do not have large hidden states that evolve continuously over time like recurrent neural networks, have emerged as powerful frameworks [30–32]. Instead of evolving hidden states, such networks include a component–dubbed an attention mechanism–that learns what part of the large sequence of inputs to focus on when computing input transformations. Moreover, variants of such architecture use hierarchical structures of attention to better match the hierarchical structure of data and discover the context in which particular patterns are informative rather than simply filtering particular patterns [33]. Although the generalized tag propagation algorithm has effectively created an implicit hierarchy, building in an explicit hierarchy of processing may provide further benefits for tasks with global information. Indeed, Poggio et al. in [34] point out that deep networks are much better suited to functions that are a "hierarchical composition of local functions" compared to shallow networks, where local refers to bounding the

number of inputs from the previous layer to each neuron. Thus, future extensions of the generalized tag propagation framework can explore explicit hierarchical organization of different processing levels and potentially have tags propagate and interact across multiple levels.

Our finding that the masked square networks were able to successfully learn the edge-connected task despite not reflecting the true connectivity structure suggests that even partial, coarse constraints correlated to the true connectivity may be sufficient. This is particularly important from an algorithmic point of view as in many real-world tasks the locality structure is unknown, particularly tasks in which connectivity is used in the more generalized sense, e.g., connecting from present to future states as in the latter part of our study. In the brain, these constraints are likely encoded genetically and implemented through shaping circuit development [35] and thus they can only be of limited precision in reflecting the diversity of environmental connectivity structure an animal may encounter. It is thus encouraging that interactions between coarse constraints and learning yield highly effective solutions, and it will be interesting to explore their interaction in more detail. Furthermore, learning this locality structure likely enables rapid switching between different tasks in the same environment and is the subject of future work.

Our shift to a generalized tag propagation regime allows many previously suggested uses of recurrent and lateral connections to be interpreted in light of local propagation of a global tag. As pointed out by Kushnir and Fusi, even without specific task structure, propagation in recurrent neural networks allows information to spread, resulting in locally-connected networks that have access to all necessary information and still perform global computations [36]. In neuroscience, lateral and recurrent connections have been hypothesized to play an important role in diverse functions such as divisive normalization [37], predictive coding [38–40], and contextual interactions such as contour detection [17, 41, 42]. Divisive normalization–perhaps the most prominent function proposed for recurrent connections in brain circuits–can be thought of as a computation that uses a global activity tag. Contextual interactions such as contour detection are naturally based on specific forms of tags (e.g., tags that trace contours), and predictive coding can be viewed as shaping the locality structure and propagation function by the average statistics of the environment. Similarly, in machine learning, the temporal use of recurrence can be seen as transforming a task that requires a global computation across time to one that can be performed locally on any time point by selectively learning and propagating a tag through time, just as the tag propagation we discuss transforms a global computation of connectivity to a local one given the input and a tag. Indeed, the internal states of RNNs performing a task can be seen as a combination of highly entangled propagated tags, and as the number of tags expand, the distinction between such networks and a generic RNN begin to blur. While this perspective does not necessarily improve our understanding of these well-established uses of recurrence, it suggests additional possible applications for more complex tag propagation. More importantly, improving interpretability is a key issue in machine learning, especially if it is to inform our understanding of neural circuits, and we believe that generalized tag-propagation dynamics offer key advantages on that front. Non-linear interactions in artificial neural networks are notoriously difficult to understand, and the addition of temporal processing in recurrent neural networks aggravates this difficulty. Indeed, complex spatiotemporal patterns and their interaction are extremely difficult to visualize and understand. In our opinion, propagation is a key exception as it can be more readily visualized and intuited. Thus, structuring networks such that they employ generalized-tag-propagation to solve tasks may lead to large advantages in terms of interpretability of the computation. Demonstrating that generalized-tag propagation can perform complex real-world tasks is an important future direction of study.

Our finding that the performance of artificial networks with hybrid architectures benefits from both the efficiency and accuracy of recurrent networks when allowed long propagation, and the effectiveness of feedforward networks when only a few steps of computation is allowed raises the question of whether neural circuits might similarly take advantage of their hybrid feedforward-recurrent architectures rather than relying simply on recurrent solutions? While neural circuits' complex architecture and temporal dynamics cannot be mapped unambiguously to fixed layer and timestep number, the following rough correspondences may prove helpful in framing comparative analyses. Layer number in artificial network simulation is intuitively related to the number of neural populations through which signals pass. Importantly, in each brain area, signals typically pass through more than one population before they reach that area's output neurons. Even assuming multiple populations per brain area along the visual pathway would have fewer than 20 processing "layers", and thus a fast feedforward pathway would be expected to produce a smaller error than a recurrent solution. The second, more subtle point is the size of the images we considered in terms of number of pixels and the length of paths within the images we used in relation to the size of the full visual field in terms of neuron receptive field size. Notably however, visual receptive field size changes with location in field of view and brain area [43, 44]. Our choice of 15x15 images was deliberately on the very conservative end to allow leeway to account for the fact that objects may not be single receptive field sized. As a point of reference, [44] estimates in primate experiments that attentional modulation in contour-tracing tasks develops slowly on the time scale of hundreds of milliseconds (propagating at 10.5 ms/degree). This timeframe, many multiples of a single neuron time constant, leaves time for multiple timesteps of propagation as assumed by our computational models.

An intriguing, related experiment is the work by Kar and DiCarlo inactivating parts of macaque ventrolateral prefrontal cortex (vlPFC) while performing an object recognition task. In brief, dividing images into those that are decodable early and late from activity in inferior temporal (IT) cortex the authors hypothesized that images that are decodable late would be more dependent on recurrent activity in vlPFC. As such, inactivating vlPFC would result in a large performance deficit for late-decodable images while having next to no deficit for early-decodable images. This result was indeed borne out in experiment. In the hybrid architectures we suggest the processing of each image could switch between being accurately processed using many recurrent steps or in an approximate feedforward pass. We therefore predict that if animals are explicitly cued to the amount of time they are allowed for processing (e.g., by performing experiments in blocks with different limitations on reaction time) late-decodable images would be shifted to approximate feed-forward processing. This would manifest as different patterns of activity associated with processing the same image under the different time constraints, reflecting the switch in processing. Alternatively, the hypothesis could be tested with more challenging perturbation experiments. Switching would predict a reduction in the effect of perturbing vlPFC even for challenging images. In both cases, the confound of overall general shorter reaction times generating changes in the dynamics needs to be addressed. We believe this confound could likely be addressed by using blocks with different restrictions on reaction time and/or using the natural variation in reaction time to generate predictions for the effect of reduced processing time without switching involved. Moreover, switching would predict a more abrupt shift when considering a range of restricted reaction times. Exploring these and other ideas for how neural circuits take advantage of their hybrid feedforward-recurrent architecture represents an exciting avenue for future work.

In summary, by extending the classical notion of connectivity as an important computation for which recurrent architectures are well suited and proposing a class of generalized tag propagation architectures to solve it, our work advanced the understanding of a crucial theoretical

question: the algorithmic role of recurrent connections. As the ability to make robust predictions in complex settings is a foundational aspect of intelligent behavior, the efficiency with which generalized tag propagation architectures solve tasks requiring temporal predictions suggests that a wider deployment of such architectures in machine learning has the potential to advance the computational capability of artificial agents, and further substantiates the idea that much of the brain architecture is geared for predicting the future state of the environment. Experimental paradigms in which animals make multiple, distinct predictions regarding future states of the environment that can be inferred from behavior or task conditions will thus be invaluable for understanding its neural implementation and deriving further constraints.

## Methods

### Code

The code for running the experiments in this paper and the associated data produced is available in the following Github repositories:

- Edge-connected pixel task https://github.com/druckmann-lab/edgeConnectedPixel

- Competitive foraging task https://github.com/druckmann-lab/competitiveForaging

### Previous networks architectures for detecting connectedness

In this section, we detail the mathematical form of the networks discussed in the introduction for detecting connectedness. In Minsky and Papert's definition of a perceptron in [1], the scalar output $y$ is assumed to be a function of $d$-predicates $\psi_1(\mathbf{I}),\ldots,\psi_d(\mathbf{I})$, i.e. d functions of the input image $\mathbf{I}$. The perceptron function is then defined by a $d$-dimensional vector $\mathbf{w}$ and threshold $b$:

$$y = \begin{cases} 1 \text{ if } \left(\sum_{i=1}^{d} w_i\, \psi_i(\mathbf{I})\right) + \mathrm{b} > 0 \\ 0 \text{ otherwise} \end{cases} \tag{6}$$

If the predicates $\psi_i$ are allowed to be arbitrary non-linear functions, then this network is a support vector machine combined with the kernel trick. The non-linear functions form a new feature space in which we learn a hyperplane boundary for classification. In practice, the predicates are defined by the user and the weights and biases are learned.

Minsky and Papert's insight was to consider how the size of the perceptron had to scale with the image size in order to perform the connectivity task when the predicates were restricted to be "local." In their definition, a predicate is local if the number of image pixels used to calculate each predicate is bounded (the bound is referred to as the order of the predicate). Minsky and Papert demonstrated that there was no set of finite order predicates which could perform the binary connectivity task as the size of the image was scaled, concluding that connectivity is a fundamentally serial task. [1] Intuitively the problem is that the connectivity between two pixels can altered by flipping any bit in the image meaning the perceptron can only perform the task if the predicates have access to all the pixels in the image.

If we limit the predicates $\psi_i$ to be linear functions followed by a shared non-linearity, then the perceptron considered by Minsky and Papert is equivalent to the modern usage of a perceptron with one hidden layer. It first computes a hidden vector $\mathbf{h}$ as a function of the input $\mathbf{x}$ according to the function $\mathbf{h} = \sigma(\mathbf{W}_1\mathbf{x}+\mathbf{b}_1)$, where $\mathbf{W}_1$ is a matrix and $\sigma$ a non-linear element-wise function. The output $\mathbf{y}$ is then computed in terms of this hidden layer: $\mathbf{y} = \sigma(\mathbf{w}_2^{\mathsf{T}}\mathbf{h} + \mathbf{b}_2)$. The weights and biases of both layers are typically learned via backpropagation. As discussed by Roelfsema [8, 9], single hidden layer feedforward networks can solve the two-pixel

connectedness task using a hidden unit for every possible path between the two pixels and the output layer returns +1 if one of these hidden units is activated and -1 otherwise. However, in order for this solution to perform the task perfectly, the number of hidden neurons must equal the number of patterns without loops that connect the two pixels and thus scales exponentially with the size of the image. (Note that if we want to detect all pixels connected to a set of points such as the edge pixels, some of the patterns can be shared but the scaling will remain exponential in the number of pixels.) Such a solution illustrates Minsky and Papert's concept of non-local computation as the longest pattern scales in length with the size of the image and thus a hidden unit potentially needs access to the whole image [1]. Concretely, consider the central pixel of the image. This pixel can be connected to the edge by a non-overlapping path that forms a spiral around this pixel. Detecting this pattern requires input from pixels throughout the image, each of which can toggle its value and change the output.

As discussed in the Introduction, a much more efficient solution is tag propagation via a recurrent architecture. This algorithm has a number of advantages over the single hidden layer perceptron. First, the number of calculations at each time step scales on the order of the number of pixels and these computations can all be performed in parallel. Second, the calculations are highly local; they require the value of the pixel and its four neighbors, a property which does not change with the size of the image. Thus, by using a recurrent vs. feedforward architecture we are able to realize Minsky and Papert's ideal of neural computing: many repeated local computations performed in parallel solve a task that is global in nature (it requires all the pixels in order to make a decision).

This solution, however, has an additional advantage that becomes evident when we consider the finding all pixels connected to the edge of the image, which we call the edge-connected pixel task. Here we are interested in finding all pixels connected a set of source points (the edge pixels) rather than a single point. Unlike the pattern matching solution which requires a significant increase in hidden units to account for the additional source pixels, the tag propagation algorithm can remain the same and perform the computations in parallel. We simply initialize the initial state to have a connected tag for all the source pixel and the recurrent network will propagate the tag in parallel, all via local computations.

In [8, 9], Roelfsema considered a pyramidal modification to the tag propagation to reduce the number of time steps required on average to converge to the correct solution. In the standard tag propagation algorithm, the required number of time steps is equal to the length of the longest path in the image for which no shorter path exists. In the worst case, this scales with the number of pixels in the image (e.g. the non-overlapping spiral from the edge to the center pixel). In the pyramidal algorithm, the first level of the hierarchy performs the tag propagation in the normal manner. In the second layer, each pixel is connected to a 3×3 square of 9 pixels in the first layer and activates only if this square has no disconnected components. In this layer the same tag propagation rules apply. This is repeated recursively until the top layer has only 1 pixel, and thus the network forms a pyramid of recurrent networks. If the pattern is such that each 3×3 grid of the image has no disconnected components, propagation can occur much more quickly through the higher layers of the pyramidal architecture. Note that this does not decrease the number of trainable weights, but rather increases the speed of propagation.

## Edge-connected pixel: Experimental set-up

Here we discuss our experimental design choices which seek to mitigate confounding factors in comparing the inherent capabilities of a given architecture to perform the edge-connected pixel task. These include the design of the sample generator, the training and hyperoptimization procedure, and the addition of skip connects to certain deep feedforward models.

For the experiments in this paper, we generated samples with a 15×15 image size (**S1 Fig**). Each pixel in the image is encoded to be off if it has a value of -1 and on if it has a value of +1. The sample generator starts with an image of all -1's and then randomly turns a set of seed pixels to +1 by independently drawing an independent Bernoulli random variable with success probability 0.05 for each pixel. For each of these seed pixels, we perform a random walk in the following manner: continue straight with probability 0.65, turn right or left with probability 0.125 each, or terminate the path with probability 0.1. The initial direction of the walk is chosen uniformly over the four cardinal directions and each path continues the walk until it terminates or hits the edge of the image.

We next seek to increase the number of distractors by randomly disconnecting paths from the edge. This is done rather than randomly turning on pixels in the image because it creates distractors with clusters of pixels that look more like paths. The procedure is defined by a cutoff and a disconnect probability. The cutoff determines how far along the path to disconnect, and the disconnect probability gives the independent success probability for disconnecting any path. For our experiments we uniformly chose the cutoff to be 0 or 1, meaning disconnecting the path meant turning off either the pixel on the edge of the image or one step away from the edge. The sample generator then finds all such pixels and turns them off with 80% probability.

The final step of the sample generator is to ensure a user-defined distribution over path lengths. For any connected pixel in the image, we can find the shortest path to the edge via on pixels. A simple measure of the complexity of a task sample is the largest such value over all pixels, i.e. what is the longest path from the edge to some pixel for which there are no other shorter paths. This property is easily calculated using the tag propagation solution: start with a connected tag for all edge pixels which are on and count the number of propagation steps until the set of labelled pixels stops changing.

To enforce the requested distribution, the sample generator first calculates how many samples of each path length it needs to achieve this distribution. For our experiments, we used a uniform distribution over path lengths from 1 to 25 so that if n samples were requested, the generator would need n/25 samples for each path length in this range. Each time a sample is generated by the random walk method, we calculate its path length and then check how many of this path length we already have. If we already have n/25, this sample is thrown out; otherwise, we keep it and increment the count for this path length. If a path length is over 25, it is counted in the quota for path length 25.

The sample generator guarantees the path distribution requested by the user, but it does not guarantee a certain number of path or distractor pixels. Generating 100,000 samples we observe the following empirical distribution over pixel types: 15.9% path pixels, 8.3% distractor pixels, and 75.8% off pixels. This achieves our goal of forcing both of the trivial solutions to the task to have high error. Labelling all the pixels as "disconnected" means an overall error of 15.9% while labelling all the on pixels as connected gives an error of 8.3% as illustrated in **Fig 2D**.

Next, we consider network hyperoptimization and training. The training loss function $\ell$ is a function of both the labelled image and the output of the network and determines what the network optimizes for during training. We use the mean squared error (MSE) between the full image outputted by the network and the correctly labelled image from our sample generation:

$(\mathbf{y}, \mathbf{y}^*) = \|\mathbf{y} - \mathbf{y}^*\|_2^2$ or the squared Euclidean distance. This loss treats performance on all pixels equally and rewards the network for outputting labels closer to the true label +1 or -1. Note that this loss function treats the task like a regression problem, but when analyzing the output, we want to evaluate the classification accuracy of the pixel. This is done in our

experiments using the sign of the network output: a positive output means the pixel is classified as +1/connected and a negative output correspond to a classification as -1/disconnected. Alternatively, one could use the binary cross entropy loss on each pixel for training, but this is not done in this paper.

All networks were implemented in Pytorch [45] and trained via its implementation of ADAM [46], a stochastic first-order optimization algorithm like stochastic gradient descent (SGD). In order to compensate for the effects of the choice of training parameters, we performed hyperparameter optimization over three parameters of ADAM for every architecture and layer combination: the learning rate, the batch size, and the weight decay. The learning rate is related to the size of steps the algorithm takes through the parameter space, the batch size is how many training examples are used in each estimate of the gradient, and the weight decay is a regularization parameter which penalizes to the $\ell_2$-norm of all the weights in the model.

The most effective method to perform hyperoptimization is still an open research question in machine learning. We used a Hyperband, a bandit hyperoptimization algorithm that begins with a large number of randomly chosen hyperparameters sets and then trains a network corresponding to each set of sampled parameters. [47] The first step is to select hyperparameter triples uniformly across a given range or equivalently selecting random points in a three-dimensional space. For learning rate and weight decay, the hyperparameters we selected uniformly over the logarithmic space, i.e. the exponent of the parameter was sampled uniformly over the range -2.5 to -5 rather than uniformly over the raw value $10^{-2.5}$ to $10^{-5}$. The batch size was sampled uniformly between 32 and 256.

For each sampled triple, we randomly initialize a network and set up an optimizer with these parameters. In random search hyperoptimization, all of these networks would be trained for a given number of epochs and then the best performing network would be selected. The insight in the Hyperband algorithm is to instead break this full training run into Hyperband epochs after which the models are pruned based on network performance. In our experiments we started out with 100 models each and used a Hyperband epoch of 1000. A given run would train the 100 models for 1000 epochs, evaluate their performance, and then eliminate the 50 models that performed the worst. The remaining models are trained for 1000 more epochs after which the networks in the bottom half of performance are again eliminated. This process is repeated until 5000 total epochs are reached, and the algorithm returns the parameters of the best performing model.

The main advantage of Hyperband over random search is computational complexity; networks in the bottom percentiles of performance are eliminated early, speeding up the subsequent epochs. Hyperoptimization, however, was not performed for the masked models, i.e. the square 5×5 mask and the neighbors mask. The first reason is that weight decay should be set to 0 for these models as the sparsity pattern in the weights has already been enforced by masking. In the fully connected model, weight decay pushes the weights towards 0 which makes sense as a regularization technique when you assume the computation has locality structure. However, when this locality structure is already encoded in the weights, the regularization can be counterproductive. Second, not performing hyperoptimization emphasizes how knowledge of the locality structure makes learning easier; it sets the masked models at a disadvantage, but we see in experiments that they still easily learn the efficient tag propagation algorithm.

Lastly, we consider the modifications made to deep feedforward networks. A common problem in such networks is the vanishing gradients in which the gradients become vanishingly small as the algorithm backpropagates through many layers [48, 49]. In our experiments, once the deep feedforward network had more than 15 layers, it got stuck across many initializations in the trivial solution that labelled all pixels as off. A common solution first proposed

**Table 1. U-Net Model Performance.** We trained four U-Net architectures of increasing size by each time doubling the number of channels in each layer. For each model, we report the best performance across 10 trained models on 10,240 test samples.

| Channels | Best Error (Path Pixels) | Best Error (Distractors) | Best Error (Path + Distractors) | Best Error (All Pixels) | Trainable Parameters | Neurons / Units |
|---|---|---|---|---|---|---|
| 2-4-8 | 5.50% | 15.76% | 9.02% | 2.23% | 2083 | 3552 |
| 4-8-16 | 0.26% | 1.87% | 0.82% | 0.19% | 8197 | 7104 |
| 8-16-32 | 0.036% | 0.12% | 0.06% | 0.011% | 32.5k | 14.2k |
| 16-32-64 | 0.0091% | 0.011% | 0.0097% | 0.0024% | 129.6k | 28.4k |

for image processing is the addition of skip connects to the network, or direct connections from early layers to later layers in the network [26]. We thus added a skip connect with fully connected weights to the input every four layers for the deep feedforward networks with 15, 20, 25, and 30 layers. Note that these skip connects take the same form as the input augmentation, and if we added a skip connect to every single layer, the two networks would be identical. Intuitively, the skip connect allows the network to reference the input image every four steps rather than every step as in the input augmented network.

## Experiments with U-Net architecture on the edge-connected pixel task

Table 1 shows the performance of the U-Net architecture [25] on the edge-connected pixel task. For these experiments, a 16×16 image was used to simplify downsampling which was performed two times to 8×8 and 4×4. The channels in the table are for each level of downsampling, i.e. the channels were doubled for each level. For counting the number of neurons, convolution followed by MaxPool or upsampling followed by convolution was considered as a single layer. All models were trained for 5000 epochs using the same hyperparameters as the input augmented models. The exact architecture used is detailed in Table 2.

We also tested the largest U-Net model on 1024 challenging images with path lengths distributed evenly between 30 and 50. Compared to the test samples with path length between 0 and 25, the error on all pixels increased from 0.0024% to 0.43% indicating that the network has not learned to perfectly perform the task. Note that tag propagation run for 50 steps achieves 0.0% error on this task.

## Details on hybrid architectures

Table 3 includes details on the hybrid architectures considered in **Fig 4C**. As shown in the figure, distribution 1 is an even split between (2, 5, 10, 30) steps of computational time,

**Table 2. U-Net Model Architecture.** U-Net architecture used for the experiments in Table 1. D specifies the base level of channels; we considered experiments with D = 2, 4, 8, 16.

| Block | Layer Type | Kernel Size | Depth | Repeat | Notes |
|---|---|---|---|---|---|
| 1 | Conv2D | 3x3 | D | ×2 | ReLU nonlinearity |
|  | Max Pool 2x2 |  | D |  |  |
| 2 | Conv2D | 3x3 | 2D | ×2 | ReLU nonlinearity |
|  | MaxPool 2x2 |  | 2D |  |  |
| 3 | Conv2D | 3x3 | 3D | ×2 | ReLU nonlinearity |
|  | MaxPool 2x2 |  | 3D |  |  |
| 4 | ConvTranspose | 3x3 | 2D |  | Concatenate with output of block 2 |
|  | Conv2D | 3x3 | 2D | ×2 | ReLU nonlinearity |
| 5 | ConvTranspose | 3x3 | D |  | Concatenate with output of block 1 |
|  | Conv2D | 3x3 | D | ×2 | ReLU nonlinearity |
| 6 | Conv2D | 3x3 | 1 |  | Tanh nonlinearity |

**Table 3. Hybrid Model Performance.** Hybrid models considered in **Fig 4C**. Models were only included in the plot if adding neurons decreased the loss. The models were assumed to be given the allowed computational time at initialization, enabling them to switch optimally.

| Model | Neurons | Path Error on Distribution 1 (Orange) | Path Error Distribution 2 (Blue) | Path Error Distribution 3 (Green) |
|---|---|---|---|---|
| Tag Propagation | 225 | 45.04% | 17.16% | 0.13% |
| Tag Propagation + 2 Layer FF | 675 | 5.49% | 3.71% | 0.13% |
| Tag Propagation + 5 Layer FF | 1350 | 29.92% | 1.59% | 0.13% |
| Tag Propagation + 2 Layer FF + 5 Layer FF | 1800 | 3.38% | 1.59% | 0.13% |
| Tag Propagation + 10 Layer FF | 2475 | 36.95% | 0.99% | 0.13% |
| Tag Propagation + 2 Layer FF + 10 Layer FF | 2925 | 4.13% | 0.99% | 0.13% |
| Tag Propagation + 5 Layer FF + 10 Layer FF | 3600 | 21.83% | 0.99% | 0.13% |
| Tag Propagation + 2 Layer FF + 5 Layer FF + 10 Layer FF | 4050 | 3.08% | 0.99% | 0.13% |

distribution 2 is an even split between (10, 30) steps of computational time, and distribution 3 always allows 30 steps of computational time.

## Competitive foraging task: Experimental set-up

To generate samples of the competitive foraging task we begin by creating a random environment (**S2A Fig**). We used 20×20 images for which -1 indicates that a pixel is part of the barrier in the environment and +1 indicates that it is open space. In the framework of the edge-connected task setup, this means that barriers are off pixels and open space are on pixels. We begin with no barriers, meaning all pixels are set to +1, before laying down between two and five rectangular barriers, the number being chosen uniformly in this range. For each rectangle, the location of the upper left corner is chosen uniformly over the pixels of the image (excluding the last three columns and rows to ensure most of the rectangle is in the image) and the width and height are chosen uniformly between 2 and 8. This procedure can result in the rectangle stretching beyond the left or bottom side of the image in which case the rectangle is cut off.

The environment now consists of several rectangular barriers which we then randomly expanded. For every pixel on the edge of a rectangle, we drew eight independent Bernoulli random variables with success probability 0.15. Each one of these corresponds to one of the eight directions one can travel from the pixel, including the four diagonal directions. For every direction that succeeds, if the pixel was not already part of the barrier we changed its value and added it to the queue to perform the same random expansion procedure. We continued this process until the queue was empty at which point the environment was finalized. Lastly, we assigned locations to the animal, competitors, and food, each of which was chosen uniformly among the open pixels. We used three competitor animals and one food location.

We next turn to the details of implementing the generalized tag propagation solution to the task. Our approach was to decompose the problems into two sub-computations: (1) generating the multi-tag propagation and (2) deciding based on the multi-tag propagation. We will show that the multi-tag propagation, as its name suggests, can be generated by a generalization of the edge-connected task and thus can be implemented via a recurrent neural network. We then designed a decision network that has access to the sequence of representations produced in a streaming fashion (i.e., one at a time, not all at once).

Consider first the animal making the decision. At every timestep $t$, we can define an $N{\times}N$ matrix $R_t$ which specifies all points in the environment that can be reached by the animal, accounting for all possible paths. The elements of $R_t$ have a one-to-one correspondence to the positions in the environment and we will use the labelling +1 for reachable at time $t$ and -1 for

unreachable at time $t$. Importantly, the range does not consider a single path, but rather all paths simultaneously.

An instructive example is to consider how the range propagates in the case of an environment with no barriers. $R_0$ will have a single reachable point: the starting point of the animal. Any point adjacent to this starting point will then be reachable in the next step, so $R_1$ will label both the starting point and its four nearest neighbors. To compute $R_t$ from $R_{t-1}$, we simply need to look at each reachable point in $R_{t-1}$ and label all of its neighbors as reachable. In this manner, for a two-dimensional grid environment with no barriers, the range will propagate out in a diamond-shaped pattern until it reached the edge of the environment. For sufficiently large $t$, the range will be the entire environment.

Now consider the modification where we add random barriers. The procedure for propagating out the range will remain the same except if a reachable point has a barrier as a neighbor, it will not be added to the range in the next iteration. The range will propagate out in the same manner, except it will stop whenever it reaches a barrier. Furthermore, the barriers may divide the grid into disconnected components where there is no way around the barrier to a certain set of points. If the environment is connected, $R_t$ for sufficiently large $t$ will be all non-barrier points in the environment; otherwise for disconnected environments, $R_t$ will converge to the set of non-barrier points in the region in which the animal starts.

We can define the same sequence of hidden representations for the competitor animals, but now instead of keeping track of the paths of one animal, we need to keep track of the paths of an arbitrary number of animals. This is simplified, however, by the fact that in our task definition we do not care which competitor is able to reach the food first. We can then track the range $R_t$ as the set of all points reachable by any of the competitors. $R_0$ will be the initial location and at each propagation in time we turn on all pixels that neighbor a reachable pixel and are not a barrier pixel. Importantly, the complexity of calculating the range does not change with the number of competitors animals; at each step, all the pixels need to check whether any of their neighbors are reachable and change their label to reachable unless they are a barrier pixel.

To implement this procedure analytically, we first observe that calculating the range for either the animal or its competitors can be viewed as a modification of the edge-connected pixel task. The environment is a set of on/off pixels, with the barriers as the off pixels. Instead of starting our tag propagation from the edges of the environment, we start from a small number of "seed" pixels that correspond either to the animal's or the competitors' starting points. We then propagate out a "reachable" label from these points to neighboring on pixels. The important difference now is we are not interested in the final labelling, but rather the sequence of labellings which tells us when a given point becomes reachable. Providing the animal locations as the initial conditions, the range propagation can be implemented by an identical recurrent network of the tag propagation algorithm as illustrated in **Fig 5B**.

From the computational point of view, calculating hidden representations in this manner also has a useful parallelism. The network can easily calculate the two ranges (one for the animal itself and the other for the competitors) at the same time since no interaction is required between the two computations. In fact, an arbitrary number of representations, each perhaps representing useful information for a decision network, can be calculated in parallel provided there are not interactions between the computations. This is the same benefit of multi-tag propagation observed for the generalized task framework.

We now turn to the decision network that receives as input the sequences generated by the multi-tag propagation, i.e., at each time step it receives the two ranges $\{\mathbf{R}_t^{\text{self}}, \mathbf{R}_t^{\text{comp}}\}$ and the location of the food pixel. In the main text, we simply concatenated these inputs together and trained an RNN on the task (**Fig 5C**). The RNN has a 25-dimensional hidden state, used the

tanh non-linearity, and learned read-out weights to a two-dimensional output. Passing this output through the softmax operator gave probabilistic belief of the network that the animal was closer to the food source (the first dimension) or its competitors were closer (the second dimension). Training was done via the binary cross entropy loss.

For comparison, we also forward engineered a recurrent network to implement what we define as the ground truth solution for the problem (S2B Fig). Consider what this input looks like for a single pixel in the environment that is not the starting location of any animal. At some time point in the competitors' range, the labelling may switch from unreachable to reachable and then will remain reachable for the remainder of the time steps. The time point at which this switch occurs indicates the length of the shortest path from the competitor that reaches this pixel given. If the point remains unreachable up until time $T$, there are two possible causes: (1) The length of the shortest path to this point is greater than $T$ or (2) the point is disconnected from the competitors in the environment in which case no path exists. In the same manner, the length of the shortest path can be extracted from the animal's own range.

Once we have the propagated tags, determining whether the animal or its competitors can reach any given point first is straightforward: whichever group has the tag which arrived earlier can reach the point first. If we perform this calculation for the food pixel, the animal should go for the food if its tag arrives first at the food and stay in place otherwise. This calculation could be performed for an arbitrary number of pixels given the same hidden representation. When specifically considering one pixel (in this case the food pixel), the network should give no preference to the animal or its competitors until one of the ranges reaches this pixel; afterwards the probabilistic belief should switch to be 1 for this group (and 0 for the other) because the network has established that this group is closer.

This decision layer can be implemented correctly using a two-neuron recurrent neural network with strong inhibitory weights (S2B Fig). Denoting these two neurons as $\mathbf{y}$, we defined the relation between decisions and $\mathbf{y}$ in the following manner: $\mathbf{y} = [1, 0]$ indicates the animal should run to the food because it is closest to it and $\mathbf{y} = [0, 1]$ indicates it should stay in place because one of the competitors is closer to the food. Denote by $\mathbf{C}$ an $N{\times}N$ matrix of zeros except for the location of the food which is denoted by a 1 (a simple conversion from the original food input which used +1/-1 to indicate the food location). At each time step, $\sum(\mathbf{R}_t^{\text{self}} * \mathbf{C})$ will be 1 if the food is reachable for the animal and -1 otherwise, given that $*$ denotes the element-wise product of two matrices and the sum is over all the elements of the results matrix. The equivalent signal for the competitors can be extracted in the same manner: $\sum(\mathbf{R}_t^{\text{comp}} * \mathbf{C})$.

Lastly, we need to add inhibition between the two neurons. We only want the neuron for either the animal or its competitors to turn on thus indicating which group was able to reach the food first; otherwise our network will output $\mathbf{y} = [1, 1]$ in the common case where the food is eventually reachable by both groups of animals in time $T$, giving no indication which group was closer. The recurrent dynamics of the decision network are then given as follows:

$$\begin{pmatrix} y_t^{\text{self}} \\ y_t^{\text{comp}} \end{pmatrix} = \sigma \begin{pmatrix} w_1 \cdot \sum(\mathbf{R}_t^{\text{self}} * \mathbf{C}) - w_1^{\text{inhibit}} \cdot y_t^{\text{comp}} \\ w_2 \cdot \sum(\mathbf{R}_t^{\text{comp}} * \mathbf{C}) - w_2^{\text{inhibit}} \cdot y_t^{\text{self}} \end{pmatrix} \tag{7}$$

The decision network thus has four learnable parameters: $w_1$, $w_2$ and $w_1^{\text{inhibit}}$, $w_2^{\text{inhibit}}$. Under the following circumstances, the network will output a correct solution to the competitive foraging task: (1) there is no sample where the food is not reachable by either the animal or its competitors, (2) for every sample, the number of time steps the network is run $T$ is larger than the shortest path to the food by any animal, (3) all the weights are positive values large enough to make the slope of the sigmoid steep (i.e. $w_1 {\geq} 10$ and $w_2 {\geq} 10$), and (4) for each neuron the inhibitory weights are larger than the weight on the network output (i.e. $w_1^{\text{inhibit}} > w_1$ and

$w_2^{\text{inhibit}} > w_2$). A larger gap in each inequality will result in a more robust system. For our implementation shown in **Fig 5E**, we used $w_1 = w_2 = 20$ and $w_1^{\text{inhibit}} = w_2^{\text{inhibit}} = 100$.

The competitive foraging task is precisely analogous to the abstract task of determining relative distance in between two groups of vertices in a graph. In each instance a subset of the vertices of the graph are randomly deleted corresponding to the barriers in the environment. There are two sets of vertices corresponding to the location of the animal and competitors respectively. The task is to return which group contains the vertex closest to a specified point in the graph, i.e. the food location. This task can be adapted to arbitrary graphs and more than two groups of nodes.

The basic CNN trained end-to-end on the competitive foraging task consisted of the following layers: Conv2D with 6 output channels followed by ReLU and MaxPool; Conv2D with 16 output channels followed by ReLU and MaxPool; three linear layers with outputs 120, 84, and 2 respectively with ReLU activations between.

## Supporting information

**S1 Fig. Sample generation procedure for the edge-connected pixel task.** The procedure for randomly generating samples of the edge-connected pixel task with a specified distribution over paths lengths and a high percentage of distractors. When used to generate a set of 50,000 samples with path lengths evenly distributed between 1 and 25, the resulting sample has approximately 15.9% path pixels and 8.3% path pixels.
(TIF)

**S2 Fig. Implementation of tag propagation via repeated Conv2D layers.** The tag propagation solution can be implemented via a sequence of repeated Conv2D layers with two input channels and two output channels. The two input channels are the current state of the tag propagation and the original input image. Output channel 1 performs one step of propagation while output channel 2 makes a copy of the original input image.
(TIF)

**S3 Fig. Experimental details for competitive foraging task.** (A) Sample generation procedure for the competitive foraging task. (B) The recurrent decision network composed without output of the propagation network in the generalized tag propagation algorithm. At each time step the network extracts whether or not the food pixel is the in range of either group of animals. Once it comes into range for one group, the corresponding neuron activates and inhibits the other neuron. The active neuron in the final time step indicates which group was closer to the food.
(TIF)

**S4 Fig. Performance comparison of all networks on the edge-connected pixel task.** Performance of the best solution after hyperoptimization for each model type across layers with results sub-divided by error type. Models with weight sharing across layers have a constant number of trainable parameters as the number of layers is varied. The circle size indicates increasing layer number. Solid colored lines connect results for a single model type as the number of layers increases. Dotted lines with arrows highlight the masked and feedforward architectures with equivalent layers or timesteps for ease of comparison Here we plot the results for all 50 runs from random initializations for each layer and architecture combinations to give a sense of the variance across runs. These runs are post-hyperoptimization and thus all use the same set of optimal hyperparameters. **S5 Fig** shows the error on all pixels, **S6 Fig** on the path pixels, and **S7 Fig** on the distractor pixels.
(TIF)

**S5 Fig. Total Error on All Runs.** For each layer/architecture combination, 50 models were trained from random initializations for 50 epochs following hyperoptimization. Here we plot the error on all pixels every 250 epochs. The gray shading indicates the range of accuracies across all instantiations.
(TIF)

**S6 Fig. Path Pixel Error on All Runs.** For each layer/architecture combination, 50 models were trained from random initializations for 50 epochs following hyperoptimization. Here we plot the error on path pixels every 250 epochs. The gray shading indicates the range of accuracies across all instantiations.
(TIF)

**S7 Fig. Distractor Pixel Error on All Runs.** For each layer/architecture combination, 50 models were trained from random initializations for 50 epochs following hyperoptimization. Here we plot the error on distractor pixels every 250 epochs. The gray shading indicates the range of accuracies across all instantiations.
(TIF)

## Acknowledgments

We would like to thank Byungwoo Kang, Minseung Choi, Tyler Benster, Winfried Denk, Jonathan Amazon and Alla Karpova for discussions and feedback.

## Author Contributions

**Conceptualization:** Brett W. Larsen, Shaul Druckmann.

**Formal analysis:** Brett W. Larsen, Shaul Druckmann.

**Investigation:** Brett W. Larsen, Shaul Druckmann.

**Methodology:** Brett W. Larsen, Shaul Druckmann.

**Software:** Brett W. Larsen, Shaul Druckmann.

**Supervision:** Shaul Druckmann.

**Visualization:** Brett W. Larsen.

**Writing – original draft:** Brett W. Larsen, Shaul Druckmann.

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
