## [Decision Letter · Decision Letter 0]

9 Dec 2021

Dear Mr. Larsen,

Thank you very much for submitting your manuscript "Towards a More General Understanding of the Algorithmic Utility of Recurrent Connections" for consideration at PLOS Computational Biology.

As with all papers reviewed by the journal, your manuscript was reviewed by members of the editorial board and by several independent reviewers. In light of the reviews (below this email), we would like to invite the resubmission of a revised version that takes into account the reviewers' comments.

We cannot make any decision about publication until we have seen the revised manuscript and your response to the reviewers' comments. Your revised manuscript is also likely to be sent to reviewers for further evaluation.

Sincerely,

Matthieu Louis

Associate Editor

PLOS Computational Biology

Kim Blackwell

Deputy Editor

PLOS Computational Biology

Reviewer's Responses to Questions

**Comments to the Authors:**

Reviewer #1: In this paper, titled “Towards a More General Understanding of the Algorithmic Utility of Recurrent Connections”, the authors explore various neural network architectures for their performance on an “edge-connected task”. They find that feedforward networks provide a “quick and dirty” solution, that is outperformed by (some) recurrent architectures, when the latter are given enough time for the signal to propagate. I furthermore find the connection to food retrieval and path-length estimates very interesting. The paper is very well written, easily accessible, and the idea is interesting. I only have few smaller comments regarding technical aspects, as detailed below.

The larger issue that I have with this paper is that I am not entirely convinced that the simplistic, and somewhat narrow pixel-connection tasks are indeed suited to provide a more general understanding of the algorithmic utility of recurrence in biological systems (as claimed in the title and paper). What the paper does provide is a modern revisit of an interesting task in which recurrence is verified to be beneficial. However, I simply do not yet see how the results have broader implications beyond this simple use case (e.g. in real-world settings and in more natural tasks). Put perhaps too simplistic: what have we learned about the role of recurrence that we did not know before? Relatedly, I am uncertain in how far the results inform us about computations performed by the brain, as quantitative comparisons with neural data or human behaviour are currently missing.

I very much hope that my comments will be perceived as constructive as they are intended and that they may help the authors improve their manuscript going forward.

Major

1. The hybrid approach feels very "engineered" based on the previous results, and I am not sure what we learn from the results given that the network architecture/behaviour is not a emergent in any one architecture. Relatedly, I am unsure in how far the results will generalise given that the hybrid solution feels “overfitted” to the task by combining separate elements that seemed to work well on the same task/dataset.

2. If I am not mistaken, the decision task was only trained on recurrent networks. Could the authors try feedforward networks on this task also?

3. The task tested here is traditionally often solved by u-net feedforward structures (I here think of the edge-connection task as a way of image segmentation). Could the authors try this architecture on their data as well?

Minor

1. “This network architecture is illustrated in Figure 2B.” Maybe Figure 2A is meant?

2. Figure 3’s caption describes 4 panels (A-D), but only A-B are shown.

Reviewer #2: Summary: The authors seek to understand the relative effectiveness of feedforward neural networks, which have been wildly successfully in machine learning, compared to recurrent neural networks, which are more difficult to train for ML applications but much closer to the architecture of real neural circuitry. The two architectures, and hybrids of them, are tested on tasks that can exploit both local and nonlocal information: i) determining whether pixels in a visual image are connected using a tag propagation algorithm and ii) generalizing this approach to a minimal model for predicting future outcomes in a competitive foraging task. One of the highlighted results is that, while both feedforward networks and recurrent networks can learn to perform these tasks, which recurrent networks being able to perform it perfectly given enough time, constraints on computation can favor feedforward networks over recurrent networks. This is because the recurrent network solutions work inward from the starting pixels, and may not converge if the pixels being queried are too far apart. The feedforward networks, on the other hand, learn a different algorithm that does not rely on tag propagation and, although not perfect, outperforms tag propagation when computation time is limited. The authors show that a hybrid network architecture can learn to use the advantages of both approaches for a given amount of computation time.

Assessment: I enjoyed the paper and have only a few clarification/discussion points that I think would improve the presentation/understanding for the reader. I otherwise support publication in PLOS CB.

Best wishes,

Braden Brinkman

Main feedback

• Fig. 3: The caption mentions panels C and D, which do not appear or are not labeled in the figure.

• P. 12: “This shows that the two network types excel in different computational regimes, the feedforward network at low computation time and the recurrent network at high computation time.” I like this summary of the authors’ result. Very clear.

• There are a few things I do not think I fully understand about Fig. 5 from the caption alone that could perhaps be clarified briefly in the caption.

--In panel B, what do the solid boundaries of the shaded diamond area represent in the last step shown? Just the boundary to where the agents can make it after a certain number of time-steps?

--Panel C: The red boundary of the yellow food pixel appears to make the pixel red unless the image is zoomed in. Perhaps remove the border? (Similarly in Fig. S2)

--Panel D: This panel refers to the learning of the propagation of tags depicted in panel B, but it appears to show only the learned propagation of the competitors, is that correct? If so, noting that this is specifically the propagation of competitor tags and the animal tag propagation is not shown would be helpful.

--Panel D: It is also not clear to me what the red error pixels represent? Are these spots that would not be marked as blue (i.e., would remain white) by the network when they should be blue?

--A legend to label the animal’s decision in panel E as ‘stay’ or ‘run’ would be helpful to the reader. In the cases where both animal and competitors can reach the food, noting that the decision depends on whichever agent reaches the food first.

---This is explained clearly in the methods, but doesn’t seem to be mentioned in the main text near Fig. 5. I think noting it in the main text would be useful as well (e.g., after “With these inputs each network was trained by stochastic gradient descent to output the correct shape of the tag at a given number of timesteps (see Methods for details).”

--I cannot parse the part of the sentence “the source pixels which being with the label”; is this a typo?

• While the competitive foraging task is of course a minimal model that does not capture a lot of complexity, it might be worth emphasizing when introduced on p. 14 that the animal has perfect knowledge of its environment (i.e., it knows with certainly the location of its competitors and the boundaries, which in a more realistic scenario may be unknown without some exploration).

Minor comments and questions

• For the analytical solution for the pixel tag propagation network (eqs. 1-2), the network weights are assumed to be the same for all neurons due to homogeneity, and different equalities are assumed to hold for the edge/corner pixels. In principle, the distance from the edges seems like it could break the homogeneity between weights, rending inhomogeneous more optimal. Could this be an alternative possibility for why the trained networks do not learn a homogeneous solution (other than gradient methods not being guaranteed to find an optimal solution, as mentioned on p. 6)? I am just asking out of curiosity, I am not sure this is an important point to address in the paper, unless the answer is interesting and worth mentioning.

• P. 18: Recommend “Such networks instead of evolving hidden states include a component, dubbed an attention mechanism” -> “Instead of evolving hidden states, such networks include a component---dubbed an attention mechanism---”

• P. 21 (under eq. 6): “then this network is a support vector machine combined the kernel trick.” -> “then this network is a support vector machine combined with the kernel trick.” (?)

• For the competitive foraging model, the authors describe an exact/analytical solution in words in the Methods; is the solution expressible mathematically? e.g., do the weights in this case satisfy a system of inequalities like in eq. 2?

**Have the authors made all data and (if applicable) computational code underlying the findings in their manuscript fully available?**

Reviewer #1: Yes

Reviewer #2: Yes

PLOS authors have the option to publish the peer review history of their article (what does this mean?). If published, this will include your full peer review and any attached files.

Reviewer #1: No

Reviewer #2: **Yes: **Braden A. W. Brinkman
---

## [Decision Letter · Decision Letter 1]

17 May 2022

Dear Mr. Larsen,

We are pleased to inform you that your manuscript '­­Towards a More General Understanding of the Algorithmic Utility of Recurrent Connections' has been provisionally accepted for publication in PLOS Computational Biology.

Best regards,

Matthieu Louis

Associate Editor

PLOS Computational Biology

Kim Blackwell

Deputy Editor

PLOS Computational Biology

Reviewer's Responses to Questions

**Comments to the Authors:**

Reviewer #1: I thank the authors for addressing all of my concerns with this submission. Congratulations on a revision well done. Signed, Tim Kietzmann

Reviewer #2: I am satisfied with the edits the authors have made to their paper in response to reviewer feedback. I enjoyed the paper and look forward to seeing it published.

I noticed just a couple of minor edits for the authors to check:

-Maybe it’s just an issue with the pdf I downloaded, but in Fig. 1F, bottom row, there are some pixels missing in the first column (compared to the original version of the manuscript, and given the ‘correct output’ image).

-P. 14: “they are comprised of” -> “they comprise” or “they are composed of”

Best wishes,

Braden Brinkman

**Have the authors made all data and (if applicable) computational code underlying the findings in their manuscript fully available?**

Reviewer #1: None

Reviewer #2: Yes

PLOS authors have the option to publish the peer review history of their article (what does this mean?). If published, this will include your full peer review and any attached files.

Reviewer #1: **Yes: **Tim C Kietzmann

Reviewer #2: **Yes: **Braden A. W. Brinkman

---

## [Editor Report · Acceptance letter]

14 Jun 2022

PCOMPBIOL-D-21-01866R1 

­­Towards a More General Understanding of the Algorithmic Utility of Recurrent Connections

Dear Dr Larsen,

I am pleased to inform you that your manuscript has been formally accepted for publication in PLOS Computational Biology. Your manuscript is now with our production department and you will be notified of the publication date in due course.

With kind regards,

Anita Estes
